



# Correction of stratospheric age-of-air derived from $SF_6$ for the effect of chemical sinks

Hella Garny[1,2], Roland Eichinger[1,7], Johannes Laube[3], Eric Ray[4,5], Gabi Stiller[6], Harald Boenisch[6], and Laura Saunders[8]

[1]Deutsches Zentrum für Luft- und Raumfahrt (DLR), Institut für Physik der Atmosphäre, Oberpfaffenhofen, Germany
[2]Ludwig-Maximilians-University Munich, Meteorological Institute, Munich, Germany
[3]Forschungszentrum Jülich GmbH, Institut für Energie- und Klimaforschung, IEK-7, Jülich, Germany
[4]Chemical Sciences Division, Earth Systems Research Laboratory, NOAA, Boulder, Colorado, USA
[5]Cooperative Institute for Research in Environmental Sciences, University of Colorado Boulder, Boulder, Colorado, USA
[6]Karlsruhe Institute of Technology, Institute for Meteorology and Climate Research, Karlsruhe, Germany
[7]Charles University Prague, Faculty of Mathematics and Physics, Department of Atmospheric Physics, Prague, Czech Republic
[8]Department of Physics, University of Toronto, Toronto, Ontario, Canada
**Correspondence:** Hella Garny (Hella.Garny@dlr.de)

**Abstract.**

Observational monitoring of the stratospheric transport circulation, the Brewer-Dobson-Circulation (BDC), is crucial to estimate any decadal to long-term changes therein, a prerequisite to interpret trends in stratospheric composition and to constrain the consequential impacts on climate. The transport time along the BDC (i.e., the mean age of stratospheric air, AoA) can best

be deduced from trace gas measurements of tracers which increase linearly in time and are chemically passive. The gas $SF_6$ is often used to deduce AoA, because it has been increasing monotonically since the 1950s, and previously its chemical sinks in the mesosphere have been assumed to be negligible for AoA estimates. However, recent studies have shown that the chemical sinks of $SF_6$ are stronger than assumed, and become increasingly relevant with rising $SF_6$ concentrations.

To adjust biases in AoA that result from the chemical $SF_6$ sinks, we here propose a simple correction scheme for $SF_6$-based

AoA estimates accounting for the time-dependent effects of chemical sinks. The correction scheme is based on theoretical considerations with idealized assumptions, resulting in a relation between ideal AoA and apparent AoA which is a function of the tropospheric reference time-series of $SF_6$ and of the AoA-dependent effective lifetime of $SF_6$. The correction method is thoroughly tested within a self-consistent data set from a climate model that includes explicit calculation of chemical $SF_6$ sinks. It is shown within the model that the correction successfully reduces biases in $SF_6$-based AoA to less than 5% for mean

ages below 5 years. Tests with using only sub-sampled data for deriving the fit coefficients show that applying the correction scheme even with imperfect knowledge of the sink is far superior to not applying a sink correction.

Further, we show that based on currently available measurements, we are not able to constrain the fit parameters of the correction scheme based on observational data alone. However, the model-based correction curve lies within the observational uncertainty, and we thus recommend to use the model-derived fit coefficients until more high-quality measurements will be



able to further constrain the correction scheme. The application of the correction scheme to AoA from satellites and in-situ
data suggests that it is highly beneficial to reconcile different observational estimates of mean AoA.

## 1  Introduction

The distributions of trace gases in the stratosphere are strongly influenced by the slow equator-to-pole overturning circula-
tion in the stratosphere, the Brewer-Dobson Circulation (BDC). Most prominently, total column abundances of stratospheric
ozone maximize in mid- to high latitudes despite strongest chemical production in the tropics. The driving of this overturning
circulation by wave dissipation, acting to drive the slow adiabatic rising and sinking as well as inducing wave stirring that
subsequently leads to mixing, has been understood for many decades (Haynes et al., 1991; Butchart, 2014). However, dynam-
ical measures of the overturning circulation like the residual circulation velocities are not observable. Rather, the distributions
of trace gases serve to deduce measures of the circulation strength - fittingly given the initial definition of the BDC as tracer
transport circulation (Brewer, 1949; Dobson, 1956).

A well-established measure to quantify the BDC strength is the mean transport time from a reference surface (e.g., the
tropical tropopause or Earth's surface) to a point in the stratosphere, referred to as mean stratospheric Age-of-Air (AoA)
(Waugh and Hall, 2002). One advantage of this AoA measure is that it can be directly deduced from trace gas abundances:
given an inert tracer whose surface abundance increases linearly with time, AoA is simply the lag time of the mixing ratio at
the surface and at a given point in the stratosphere. Strictly, this measures both the transport time within the troposphere and in
the stratosphere, but as transport times from the surface to the tropical tropopause can be neglected compared to stratospheric
transport times, AoA relative to the surface is still a good measure of the stratospheric transport circulation. However, such
an ideal AoA tracer does not exist in reality. One gas that has often been used to derive AoA from observations is sulfur
hexafluoride ($SF_6$). $SF_6$ is a well-suited AoA tracer since its surface abundances have increased monotonically since the
1950s, and its chemical sinks are located in the mesosphere and above, which previously have been assumed to have little
influence on $SF_6$ abundances outside the winter polar vortices. A number of high-quality measurements of $SF_6$ from different
platforms (in-situ, satellite) exist from many decades ago up until today (e.g. Andrews et al., 2001; Engel et al., 2009; Stiller
et al., 2012; Haenel et al., 2015). While $SF_6$ surface abundances do not increase linearly, it has been shown that the non-linear
increase in $SF_6$ can be corrected for in the derivation of AoA (Stiller et al., 2012; Fritsch et al., 2019). $SF_6$ has the advantage
compared to the other common age tracers $CO_2$ that it has almost no seasonal cycle in emissions. The strong seasonal cycle
in $CO_2$ complicates the AoA calculation in particualr in the lower stratosphere (e.g. Andrews et al., 2001). Thus, $SF_6$ is, in
principle, a useful and often measured AoA tracer. However, it became increasingly clear in recent years that the chemical sinks
of $SF_6$ introduce substantial biases in AoA (e.g. Stiller et al., 2012; Ray et al., 2017; Kovács et al., 2017; Leedham Elvidge



et al., 2018; Loeffel et al., 2022). It would be strongly desirable to be able to correct for the effects of chemical sinks in $SF_6$ in order to use the existing and possible future observations of $SF_6$ to monitor the stratospheric transport circulation.

Loeffel et al. (2022) investigated the effects of chemical sinks on AoA derived from $SF_6$ in a global model, and showed that the bias of AoA induced by the chemical sink grows with increasing mixing ratios. Thus, biases are particularly strong since the 2000s. Furthermore, this effect can induce apparent positive trends in AoA, hampering efforts to deduce long-term

circulation trends from $SF_6$-based AoA. In Loeffel et al. (2022) it was discussed that there is a compact relationship between ideal and $SF_6$-based apparent AoA for AoA below a certain threshold. On this basis, it was suggested that it might be possible to correct observed $SF_6$-based AoA for the effects of chemical sinks. Here, we follow up on this idea and develop a correction scheme for $SF_6$-based AoA. First, theoretical considerations with simplifications are used to deduce a possible formulation of the correction scheme (Sec. 2). The strategy of the paper is to develop and test the correction scheme thoroughly within the

self-consistent model world (Sec. 4, with the model being described in Sec. 3.1). This includes sub-sampling the model data to test the robustness of the correction against incomplete knowledge of the ideal to apparent AoA relationship (Sec. 4.3). The next step is a comparison of the relation of $SF_6$-based AoA to AoA deduced from other tracers from observational data to the model relation. Ideally, the fit parameters for the correction scheme would be based on observational data alone; however we show that current observations are not able to constrain the fit parameters (Sec. 5). Instead, we use fit parameters from the

model and apply them to independent observations to test the performance of the correction scheme (Sec. 5). In Sec. 6 we summarize our findings, discuss the shortcomings and give recommendations on the application of the correction scheme.

## 2 Theoretical considerations for sink correction methods

In this section, we will lay out the theoretical basis for how chemical depletion of a tracer affects the derivation of mean AoA from this tracer. We will make various simplifying assumptions to derive analytical expressions for the effects of chemical

sinks on tracer abundances and on AoA derived from it. The climate model data in the next sections will serve to test how well justified those assumptions are.

Generally, for any location we can express the stratospheric tracer mixing ratio $\chi_s$ at time $t$ for a tracer that is transported from a reference surface as:

$$\chi_s(t) = \int_{t'=0}^{\infty} \chi_0(t-t') exp(-t'/\tau(t')) G(t') dt' \tag{1}$$

where $t'$ denotes the transit time, $\chi_0(t)$ is the time-series of the tracer mixing ratio at a reference surface, and $G(t')$ represents the Green's function and is equivalent to the distribution of transit times (often called the age spectrum). The exponential function in the integral represents the effects of chemical sinks on the tracer with transit-time dependent and path-integrated lifetime $\tau$. For simplicity, we will first consider a tracer with linear increase at the reference surface, before moving on to a tracer with monotonic, but non-linear increase.






**Linearly increasing tracer**

We assume a linearly increasing tracer with $\chi_0 = \delta_{\chi_0} * t$ and further assume a single average path, i.e., that the AoA spectrum would be a delta-function (as was discussed already in Loeffel et al., 2022). The latter assumption is certainly only a rather crude approximation, but allows to make analytical progress. Under those assumptions, Eq. 1 simplifies to:

$$\chi_s(t) = \delta_{\chi_0} exp\left(-\frac{\Gamma}{\tau_{eff}}\right)(t-\Gamma) \tag{2}$$

where $\Gamma$ is the transit time of the assumed single path, equivalent to the mean AoA. The lifetime $\tau_{eff}$ is the path-averaged lifetime along the hypothetical single path. Thus, overall this lifetime can be seen as lifetime averaged along each individual path and across all the different paths; in the following we refer to this as the "effective" lifetime.

AoA for a linearly increasing tracer is given simply by the lag time between the tracer mixing ratio at the location of interest and the time-series at the reference surface. We will refer to such estimated AoA without taking effects of chemical sinks into account as the apparent AoA $\tilde{\Gamma}$. In this case, the apparent AoA is given by

$$\tilde{\Gamma} = t - \frac{\chi_s(t)}{\delta_{\chi_0}} = t\left[1 - exp\left(-\frac{\Gamma}{\tau_{eff}}\right)\right] + exp\left(-\frac{\Gamma}{\tau_{eff}}\right)\Gamma \approx \Gamma\left(1 + \frac{t}{\tau_{eff}}\right), \tag{3}$$

where the latter expression has been approximated assuming that $\Gamma$ is small compared to the lifetime $\tau_{eff}$ (i.e, $exp\left(-\frac{\Gamma}{\tau_{eff}}\right) \approx 1 - \frac{\Gamma}{\tau_{eff}}$). Thus, this expression states that the apparent AoA differs from the true AoA $\Gamma$ by factor $(1 + \frac{t}{\tau_{eff}})$, which increases as time progresses (due to the increasing tracer mixing ratios, see discussion in Loeffel et al. (2022)). The other necessary parameter for estimating the effects of chemical sinks on AoA is the effective lifetime. Since chemical sinks of $SF_6$ are located in the upper stratosphere and mesosphere, the effective lifetime is location dependent. For the correction scheme proposed here, we will parameterize the effective lifetime as a function of AoA. The effective lifetime can be calculated from simultaneous data of $SF_6$-derived apparent AoA and AoA based on another tracer, that is unaffected by the chemical sinks, via the relation derived above, i.e.:

$$\tau_{eff}(\Gamma) = t\left(\frac{\tilde{\Gamma}}{\Gamma} - 1\right)^{-1}. \tag{4}$$

We deduce this relationship based on model data and then parameterize the effective lifetime as a function of ideal AoA ($\Gamma$) via different fit functions (exponential, polynomial), and the estimated parameters then serve as input for the correction scheme.

**Non-linearly increasing tracer**

In a next step, we extend the formulations given above to correct for a non-linearly increasing tracer, as is the case for $SF_6$. We keep the assumption of a delta-function (average-path) AoA spectrum as above. Given this assumption, AoA is still calculated from a passive tracer as the time-lag between the tropospheric time-series $\chi_0$ and the timeseries at a given point in the stratosphere, with $\chi_s^p(t) = \chi_0(t-\Gamma)$.

To account for the non-linear increase in the reference time series, we linearize the tropospheric timeseries around a time $t - \Gamma_0$, so that:

$$\chi_s^p(t) = \chi_0(t-\Gamma) \approx \chi_0(t-\Gamma_0) + \frac{\partial\chi_0(t-\Gamma_0)}{\partial t}\cdot(\Gamma-\Gamma_0) \tag{5}$$





Solving the above equation for $\Gamma$ then yields:

$$\Gamma = \frac{\chi_s^p(t) - \chi_0(t - \Gamma_0)}{\frac{\partial \chi_0(t - \Gamma_0)}{\partial t}} + \Gamma_0. \tag{6}$$

Note that this is not the common way to account for non-linearity when deriving AoA from a non-linear increasing tracer.

Rather, either a convolution with the age spectrum or a second order polynomial fit to the reference time-series is used (Engel et al., 2009; Stiller et al., 2012; Fritsch et al., 2019). The simple linearization of the tropospheric reference time-series only approximates the local slope. For the mean age calculation, the time-evolving slope over all transit times that contribute to the age spectrum is important. However, for the sink correction we find that the approximation of linearizing around a certain mean age proves to work well (as shown in the following in Sec. 4.2).

To obtain the relationship of apparent age to the true ideal age, we insert the chemically depleted tracer mixing ratio $\chi_s(t)$ instead of the passive tracer in Eq. 6. We assume a delta-function AoA spectrum as before and thus can insert $\chi_s(t) = \chi_0(t - \Gamma) \cdot exp(-\Gamma/\tau_{eff})$ (from Eq. 1). Furthermore, based on the linearization of the reference time-series above, we define a time-dependent function $F_t(t)$ as the ratio of the reference mixing ratio to its temporal gradient:

$$F_t(t) = \frac{\chi_0(t - \Gamma_0)}{\frac{\partial \chi_0(t - \Gamma_0)}{\partial t}} \tag{7}$$

This time-dependent function will be described in more detail below. With inserting those expressions for $\chi_s(t)$ and $F_t(t)$ in Eq. 6, we obtain:

$$\tilde{\Gamma} = exp\left(-\frac{\Gamma}{\tau_{eff}}\right) \cdot (F_t(t) + \Gamma - \Gamma_0) - F_t(t) + \Gamma_0. \tag{8}$$

Further using the approximations of small AoA versus lifetime as above we can approximate $exp(-\Gamma/\tau_{eff}) \approx (1 - \Gamma/\tau_{eff})$, yielding:

$$\tilde{\Gamma} \approx \Gamma\left(1 + \frac{F_t(t)}{\tau_{eff}}\right) + \frac{\Gamma\Gamma_0}{\tau_{eff}} - \frac{\Gamma^2}{\tau_{eff}} \approx \Gamma\left(1 + \frac{F_t(t)}{\tau_{eff}}\right) \tag{9}$$

The last step follows from assuming $\Gamma_0 \approx \Gamma$. Thus, we obtain a similar expression as for the linearly increasing tracer (see Eq. 3), but replacing the linear time-dependence $t$ with the time-dependent function $F_t(t)$. If the reference tracer mixing ratio increases linearly, Eq. 9 collapses to Eq. 3.

**Correction scheme**

The correction scheme we propose here is based on Eq. 9, and contains the following steps:

1. For any pair of given data of apparent and ideal AoA ($\tilde{\Gamma}_i$, $\Gamma_i$) observed at a certain time $t_i$, the effective lifetime is calculated based on $\tau_{eff} = F_t(t_i)/(\tilde{\Gamma}_i/\Gamma_i - 1)$.

2. A function is fitted to parameterize $\tau_{eff}$ as a function of ideal AoA ($\Gamma$). In Sec. 4.1, we test an exponential fit function as well as polynomial fit functions of different order. The fits are performed over a limited range of ideal AoA values (here 140 1 to 5 years, see Sec. 4.1).





3. Given the parameters from the fit, and the time-dependent function based on the reference time-series $F_t(t)$, the relation between $\tilde{\Gamma}$ and $\Gamma$ is given for any time and can be used to correct $SF_6$-based apparent AoA. Since the relation between $\tilde{\Gamma}$ and $\Gamma$ cannot be analytically inverted (due to the non-linear dependence of $\tau_{eff}$ on $\Gamma$), we in practice calculate $\tilde{\Gamma}$ for a range of ideal mean ages, and $\Gamma$ is estimated from $\tilde{\Gamma}$ via linear interpolation. AoA values beyond the range for which the fit was performed in the previous step (here for ideal AoA > 5 years) are treated in the following way: in case of the exponential fit, the exponential fit function is evaluated also beyond AoA > 5 years. In case of the polynomial fit, this would lead to problems as the polynomial is not well constraint outside the fit range, and does not increase monotonically. Therefore, the polynomial is not evaluated for AoA > 5 years, but values are extrapolated linearly.

**Time-dependent function for non-linear increase**

The time-dependent function $F_t(t)$ takes the non-linear increase of the reference mixing ratio timeseries into account and is defined as the ratio of the mixing ratio and its temporal slope at a given time as given by Eq. 7.

The reference $SF_6$ time-series at the surface is shown in Fig. 1 (top), both for the best estimate based on surface observations (orange) and the time-series taken from the global model data (blue). As can be seen, the two time-series are close to identical. Nevertheless, in the remainder of the paper the model time-series is used for model $SF_6$ data, and the observational time-series is used for correcting observed $SF_6$-based AoA (in Sec. 5). The middle panel shown the temporal gradient in the reference time-series. $F_t(t)$ (bottom panel) is calculated based on the annual mean time-series of $SF_6$, where the gradient in the denominator of $F_t(t)$ is calculated as centered difference over five years. We find that in practice the offset mean AoA ($\Gamma_0$) of 4 years gives best results, but generally results were found to be robust with respect to small variations in the choice of the offset mean AoA and also of the period for the gradient calculation. The values of $F_t$ from the model versus observational time-series only show differences in the earlier years (1950s, and less so in 1970s), which have negligible effects on the results presented in the remainder of the paper.

## 3 Data and Model description

### 3.1 EMAC time slice simulation

We use the ECHAM/MESSy Atmospheric Chemistry (EMAC) model, which is a numerical chemistry and climate simulation system that includes submodels describing tropospheric and middle atmospheric processes and their interaction with oceans, land and human influences (Jöckel et al., 2010). It contains of the 5th generation European Centre Hamburg general circulation model (ECHAM5, Roeckner et al., 2006) as dynamical core atmospheric model and uses the second version of the Modular Earth Submodel System (MESSy2) to link multi-institutional computer codes. The physics subroutines of the original ECHAM code have been modularized and reimplemented as MESSy submodels and have been continuously further developed.

For the present study we applied EMAC (MESSy version 2.54.0; Jöckel et al., 2010, 2016) in the T42L90MA-resolution, i.e. with a spectral truncation of T42, which corresponds to a quadratic Gaussian grid of ~2.8° x ~2.8° in latitude and longitude and with 90 hybrid pressure levels in the vertical that reach up to 0.01 hPa. The applied model setup comprises the basic submodels

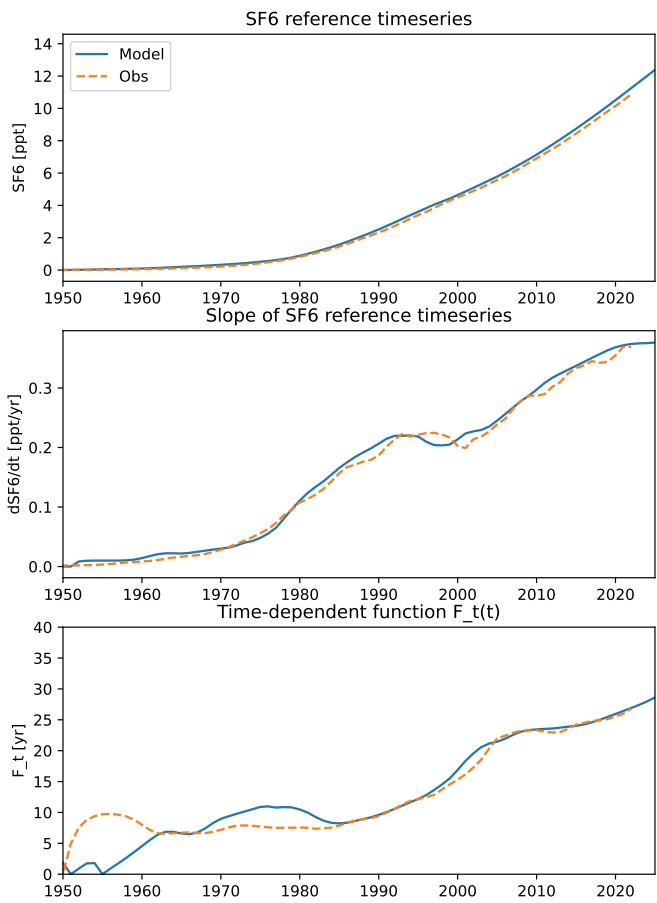

**Figure 1.** Top: Reference timeseries of $SF_6$ mixing ratios at the surface based on observations (orange dashed) and taken from the lowest level of the global model (blue). Middle: the temporal gradient of the two timeseries calculated as as centered difference over five years. Bottom: the time-dependent function $F_t(t)$ defined as the ratio of the mean to the gradient, and shifted by $\Gamma_0$=4 years.

for dynamics, radiation, clouds, and diagnostics (AEROPT, CLOUD, CLOUDOPT, CVTRANS, E5VDIFF, GWAVE, ORBIT, OROGW, PTRAC, QBO, RAD, SURFACE, TNUDGE, TROPOP and VAXTRA (see Jöckel et al., 2005, 2010, for details))

and the submodel SF6. The submodel SF6 is used to explicitly calculate the chemical sinks of $SF_6$ in the middle atmosphere. The calculations are based on the reaction scheme by Reddmann et al. (2001), details of the submodel are provided in Loeffel et al. (2022), but in the following we summarize the most important points: For electron attachment-based $SF_6$ degradation, the electron field by Brasseur and Solomon (1986) is used and UV photolysis of $SF_6$ is not included as its loss rate is several orders of magnitude weaker than that of electron attachment up to about 100 km. Further reactions considered are photodetachment

of $SF_6^-$ (Datskos et al., 1995), destruction of $SF_6^-$ by atomic hydrogen, hydrogen chloride, and ozone (Huey et al., 1995), stabilization of $SF_6^-$ excited by collisions as well as autodetachment of $SF_6^-$. The autodetachment rate was set to $106\,\mathrm{s}^{-1}$ and



to compute the photodetachment rate of $SF_6^-$, we use the extraterrestrial solar photon flux with no attenuation of the UV-photon flux (see Reddmann et al., 2001), as provided by WMO (1986).

The simulation setup is designed as a time slice simulation with climate conditions of the year 2000. This means that climatologies of the 1995–2004 period of the greenhouse gases (GHGs) $CO_2$, $CH_4$, $N_2O$, and $O_3$, the sea surface temperatures (SSTs) and the sea ice concentrations (SICs) are prescribed as monthly means. For the GHGs, the ESCiMo RC1-base-07 simulation (see Jöckel et al., 2016) and for SSTs and SICs, the Hadley Centre Sea Ice and Sea Surface Temperature (HadISST) dataset (which was also used for the RC1-base-07 simulation) were taken. The CCMI-1 volcanic aerosol data (for its effect on infrared radiative heating, see Arfeuille et al., 2013; Morgenstern et al., 2017) are prescribed and the quasi-biennial oscillation (QBO) is nudged (see Jöckel et al., 2016). Moreover, the 1995–2004 average of the reactant species for the $SF_6$ sinks, namely HCl, H, $N_2$, $O_2$, $O(^3P)$, and $O_3$ from the RC1-base-07 simulation have been prescribed. Other than the SF6 submodel, no interactive chemistry is activated in the simulations for this study. This simulation has also been used in Loeffel et al. (2022) and therein was referred to as the TS2000 simulation. The setup is designed to investigate the effects of the $SF_6$ sinks under constant climate conditions. Note, however, that in contrast to the climate conditions, the $SF_6$ lower boundary conditions are prescribed transiently over the period of the simulation with values starting in year 1950.

We use the same AoA tracers as described in Loeffel et al. (2022), in particular the linearly increasing ideal AoA tracer and two $SF_6$-tracers, one without chemical depletion and one with chemical depletion. AoA is calculated from the tracer mixing ratios as described in Loeffel et al. (2022), following Fritsch et al. (2019). In brief, we correct for the non-linear increase in the reference time-series in the calculation of AoA from the $SF_6$ tracers via a polynomial fit method (with the parameters of ratio of moment of 1.0 and fraction of input 95%; see Fritsch et al. (2019) for details). With this method, the passive $SF_6$ tracer results in almost identical AoA than the ideal, linearly increasing AoA tracer. In the following, we use the two $SF_6$ tracers with and without sinks and refer to AoA from $SF_6$ without sink to "ideal" AoA and to AoA from $SF_6$ with sink to "apparent" or $SF_6$-based AoA.

## 3.2 Observational data

### 3.2.1 Aircraft and balloon data

Concurrent measurements of $SF_6$ and other, alternative age tracers are necessary to constrain the ideal to apparent AoA relationship based on observations. Here, we use a number of such measurements available from aircraft and balloon campaigns spanning a period of 21 years (1997 to 2017) taken during different seasons and at different latitudes (see Fig. 2 for an overview).

Leedham Elvidge et al. (2018) described stratospheric observations of a set of inert trace gases and demonstrated their usefulness as AoA tracers, including a comparison to $SF_6$-based AoA. The five species are $CF_4$, $C_2F_6$, $C_3F_8$, $CHF_3$ (HFC-23), and $C_2HF_5$ (HFC-125), all of which result in slightly lower AoAs as compared to $SF_6$, with the discrepancy increasing with increasing AoA (see Fig. 2). This data set is based on air samples collected during several high altitude balloon (labelled



B34 and B44 in Fig. 2) and aircraft (labelled OB09, K2010, K2011, SC2016 in Fig. 2) campaigns in the tropics, mid, and high
latitudes. It was subsequently augmented by AoA data from a further aircraft campaign (SC2017, Adcock et al., 2021).

The balloon data from 1997, 1998 and 2000 come from in situ measurements of $SF_6$ and $CO_2$ taken during the Observations
of the Middle Stratosphere (OMS) and SOLVE campaigns (Volk et al., 1997; Andrews et al., 2001; Moore et al., 2003; Ray
et al., 2017). The mean ages were calculated differently for each trace gas although they both use a variation of the lag
technique. The sampling rates were also different between the instruments measuring each trace gas so the points shown in
Fig. 2 for these flights were chosen based on a time coincidence of +/- 30 seconds.

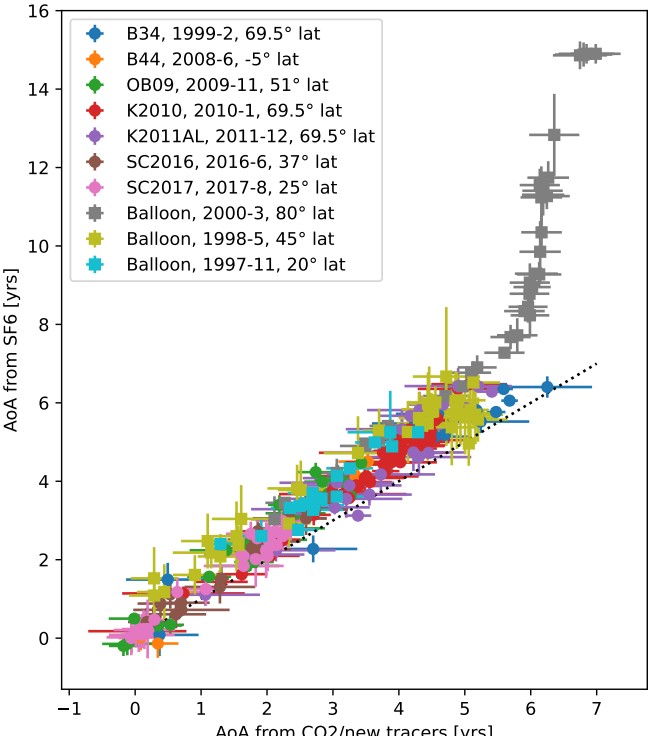

**Figure 2.** Available observational data pairs of AoA from $SF_6$ (y-axis) and AoA from $CO_2$ (for the three Balloon flights) or the average
AoA from up to five alternative age tracers (x-axis). Data are taken from different balloon and aircraft campaigns (see legend including
information on year-month, and latitude of observation; see text for details). Error bars represent one standard deviation and the dotted line
is the 1:1 line.

### 3.2.2  Satellite data

MIPAS was a FTIR spectrometer on the Envisat satellite. It scanned the atmosphere from July 2002 to April 2012 in an al-
titude range between 6 and 70 km. During this period, about 2 million $SF_6$ profiles, covering the entire globe and measured
during day and night have been obtained. The retrieval of $SF_6$ profiles from the radiance spectra has been described previ-





ously (Stiller et al., 2008, 2012; Haenel et al., 2015). The data version used here has been retrieved following the approach
described by Haenel et al. (2015), however, using newer absorption cross sections of $SF_6$ (Harrison, 2020) and accounting for
a trichlorofluoromethane (CFC-11) band in the vicinity of the $SF_6$ signature (Harrison, 2018). This $SF_6$ data record is publicly
available at https://doi.org/10.5445/IR/1000139453 (Stiller et al., 2021).

Moreover, we use AoA deduced from $N_2O$ from the GOZCARDS (Global OZone Chemistry And Related trace gas Data
records for the Stratosphere, (Froidevaux et al., 2015)) satellite product, as produced by Linz et al. (2017). The $N_2O$ data are
available for the years 2004–2012 and based on observations by ACE-FTS (for 2004–2010), and by Aura MLS (2004-2012).
The AoA product relies on an empirical relation between $CO_2$-based AoA and $N_2O$ as found by Andrews et al. (2001), with
accounting for the tropospheric trend in $N_2O$ (Linz et al., 2017).

## 4   Testing the correction scheme in a global model

In this section, we use self-consistent model data to develop a sink correction scheme based on the concept laid out in Sec. 2.
We start with an overview of the simulated relation between ideal and $SF_6$-based apparent AoA. Fig. 3 shows the difference
between ideal AoA and apparent AoA derived from the chemically depleted $SF_6$-tracer as a function of ideal AoA. As found
by Loeffel et al. (2022), AoA differences due to chemical sinks increase approximately linearly for AoA lower than about 4
years, and increase very strongly for older AoA values. The AoA difference increases from the 1960s to the 2020s due to the
increasing mixing ratios of $SF_6$ (see Fig. 3 left). The increase in differences is smaller in the first decades, when the $SF_6$ growth
rates are small (see Fig. 1), and differences increase more strongly with the stronger growth rates after  1990. The seasonal
and latitudinal variations in the AoA differences are highlighted in the right panel of Fig. 3 for one specific year (2015). The
ideal to apparent AoA relation is generally found to be quite compact for AoA below about 4 years. Variations with latitude
and season are found in the ideal AoA values for which the difference starts to strongly increase: the strong increase of the
AoA difference is found at lower ideal AoA values for tropical air compared to high latitudes. The contrast of tropical to high
latitude air will be discussed in more detail in the next section. Generally, the mostly compact relationship between ideal to
apparent AoA is promising for our aim to deduce a correction method that is valid globally.

Our strategy in developing the correction scheme is as follows. The first step is to parameterize the effective lifetime as
function of AoA by using different fit functions (exponential or polynomial, see Sec. 4.1). We put particular focus on testing
whether the fit coefficients change over time, and how strongly they vary with latitude. Note that we have tested all shown
diagnostics first with a pair of linearly increasing tracers (one with and one without chemical sink), but since we found very
similar results for the non-linearly ($SF_6$-like) increasing tracer, we only show results for the latter more realistic case. We then
apply the correction schemes with fit coefficients from the model to the model data itself (Sec. 4.2) to quantify by how much
the biases in $SF_6$-derived apparent AoA can be eliminated by the sink correction. This serves to decide which fit function
performs best, and how much the results improve through latitudinal dependent fit coefficients compared to global uniform fit
coefficients. Finally, in Sec. 4.3, it is tested how limited availability of data in space and time impacts the fit coefficients and
the subsequent correction, as will be relevant for the application to observational data.





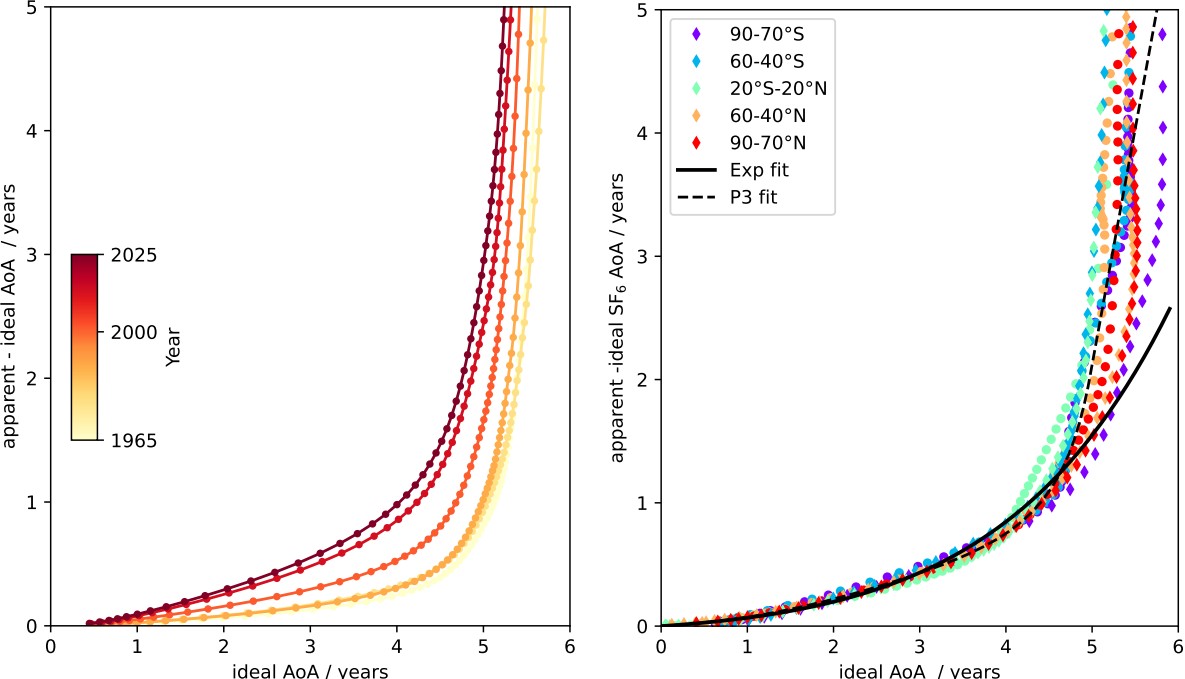

**Figure 3.** Left: Mean AoA from the ideal tracer plotted against the difference of apparent mean AoA derived from the non-linearly increasing $SF_6$-like tracer to ideal mean AoA from the TS2000 model simulations, for global and decadal averages from year 5-15 (equivalent to 1965-75 in $SF_6$ emissions, yellow) to year 55-65 (equivalent to 2015-2025, dark red). Right: as left, but values for individual latitude bands and seasons for year 2015 with circles denoting DJF and diamonds JJA, color coded by latitude band (see legend). The black fit lines are the global mean fits used for the correction, see Sec. 4.1 for details.

## 4.1 Fit coefficients for different fitting methods

We first present two exemplary profiles of apparent AoA and ideal AoA, one in the tropics (10°S-10°N) and one over the south pole (80-90°S) arbitrarily chosen for the year 2000. In Fig. 4, the difference of apparent to ideal AoA is shown as function of ideal AoA (left panel; similar to Fig. 3, but restricted to lower AoA values), showing the strong, non-linear increase of the AoA difference with increasing ideal AoA. For ideal AoA up to about 4 years, the difference between apparent and ideal AoA is lower for tropical than for polar latitudes. This can be understood from a lower fraction of old air in the tropics compared to the high latitudes for a given AoA, i.e., less weight of the tail of the AoA spectrum (see e.g. Ploeger and Birner, 2016). The older air is more likely to be depleted in $SF_6$. For AoA above about 4 years, on the other hand, the difference between apparent and ideal AoA is larger in the tropics compared to high latitudes. Ideal AoA of 4 years is located at a pressure level of around 7 hPa in the tropics, but as low as around 60 hPa in high latitudes. Thus, the tropical air with ideal AoA above about 4 years is located much higher in the atmosphere, and therefore closer to the region where chemical sinks are active. This likely explains the strong increase of the apparent AoA already for lower ideal AoA compared to high latitudes. Next to the annual mean profiles,





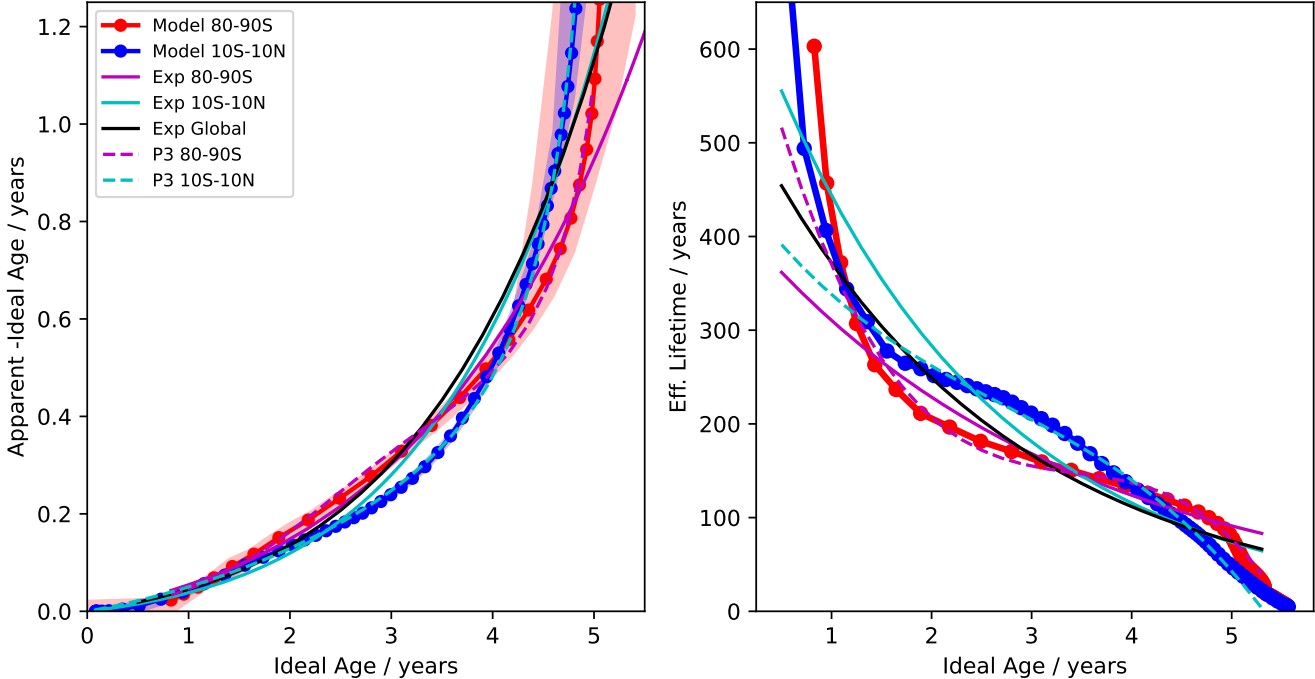

**Figure 4.** Left: Difference in apparent to ideal AoA for annual mean profiles of year 2000 in the tropics (blue) and southern high latitudes (red) as a function of ideal mean age. The range of monthly mean values are shown as shaded regions (red for southern high latitudes and blue for tropics). Right: Effective lifetimes $\tau_{eff}$ calculated from the relationship $\tilde{\Gamma} = \Gamma \left( 1 + \frac{F_t(t)}{\tau_{eff}(\Gamma)} \right)$, for the two profiles together with different fits of $\tau_{eff}$ with exponential (Exp; solid) and 3rd order polynomials (P3; dashed) to the tropical profiles (cyan), the southern high-latitude profile (red) and to global data (black). See legend on left for both panels.

shaded regions display the variation over the year. The seasonal cycle in the ideal-apparent AoA relation is very small in the tropics (see light blue shading in Fig. 4, left), whereas it is more pronounced at high latitudes for AoA values above about 4 years. Specifically, the AoA difference for a given ideal AoA value rises strongly in early spring (around November; not shown). This sudden increase in apparent AoA for ideal AoA of around 5 years is due to a sudden rise in the AoA isosurface: while the 5 year contour is located at around 30 to 40 hPa in winter, it rises in altitude to above 10 hPa in spring (not shown),

thus we are looking at a different air mass. If we analyse the apparent to ideal AoA difference on a fixed pressure level (e.g. 20 hPa), this difference is highest in late winter just before the polar vortex breaks down (October), and strong downwelling accumulated $SF_6$-depleted air over winter in the stratosphere. However, apart from spring at high-latitudes, the relationship between ideal and apparent AoA does not have a strong seasonal cycle (see also Fig. 3), and we will consider only annual mean profiles for performing the fits hereafter.

The effective lifetimes for the tropics and high-latitudes calculated from the ideal and apparent AoA profile via Eq. 4 are shown in the right panel of Fig. 4, and reflect the behavior observed in the AoA differences. For both profiles, the lifetime of





$SF_6$ in young air is very long, and it drops rapidly for old AoA. For AoA between about 1 to 4 years, the effective lifetime of tropical air is longer as compared to high latitudes, but it drops to shorter effective lifetimes for ages above about 4 years. The values of the effective lifetimes lie around 200 years. As noted in Sec. 2, the effective lifetime is the path-integrated lifetime

and should not be confused with the total middle atmospheric (stratospheric+ mesospheric) lifetime of $SF_6$. The latter was estimated to be about 1900 years in the simulation used here (Loeffel et al., 2022).

For the sink correction scheme, a fit to parameterize the effective lifetime as function of AoA is sought. The fit aims at ages between about 1 and 5 years ideal AoA. Below 1 year, the effects of chemical sinks are small (strong increase in effective lifetime, see Fig. 4 right), and for air with AoA above 5 years, the relationship between ideal and apparent AoA gets very steep

(see also Fig. 3). Therefore, we perform the fits of $\tau_{eff}$ for the range of AoA between 1 to 5 years. Tests with variations of this range showed that performing the fits for this interval gives generally best results in terms of the bias correction of apparent $AoA$. For the AoA range between 1 to 5 years, $\tau_{eff}$ roughly decreases exponentially with AoA. Thus, we test an exponential function as fit of $\tau_{eff}(\Gamma)$. However, in particular in the tropics the turning point in the $\tau_{eff}$-AoA relationship around 3 years cannot be captured by an exponential function, and we also test a 3rd order polynomial fit to capture these turning points better.

As shown by the fit lines, the exponential fit captures roughly the decrease of lifetimes with AoA, but the 3rd order polynomial is better suited to capture the curvature of $\tau_{eff}$. We further tested a 5th order polynomial fit, which is only slightly superior to the 3rd order fit (not shown in Fig. 4) at the cost of more fitting parameters.

The resulting apparent - ideal AoA differences using the fit functions for the effective lifetime are shown as solid and dashed lines in the left panel of Fig. 4. Note that while the fit was performed only for 1 years < AoA < 5 years, the functions are still

evaluated for lower AoA values. For AoA > 5 years, the exponential function is as well evaluated. For the polynomial fit, the effective lifetime approaches zero for high AoA values, so that the apparent AoA would go to infinity, and switch sign after the zero crossing of the effective lifetime. This is only a problem for ideal AoA above 5 years, i.e., values above the range the fits were performed for. To avoid this nonphysical behaviour, the apparent AoA for ideal AoA above 5 years is extrapolated linearly instead of using the polynomial fit function (see Sec. 2). The effect of this can be seen in Fig. 3 (right) panel, and leads

to a rather more conservative correction for high AoA values.

Consistent with the fits to the effective lifetime, the polynomial fits capture the AoA differences better, in particular for older AoA. The strong increase in the apparent - ideal AoA difference for older air is not fully captured by the exponential fits, however they do still capture the general increase in the apparent to ideal AoA difference reasonably well. When using global fit coefficients, obtained by fitting $\tau_{eff}$ values from all latitudes simultaneously, the general shape of the apparent to ideal AoA

difference can be reproduced well (as shown for the exponential fit in Fig. 4 as black line).

Overall, the deviations between the fit lines for different regions or different fit functions are small (< 0.1 years) for AoA below 5 years, indicating that the specific details of the fit might be of little importance for the AoA correction. A more quantitative assessment of the correction with the fitted $\tau_{eff}$ functions for all latitudes will be given in the next section.

As discussed in Sec. 2, by including the time-dependent function $F_t(t)$ the lifetimes ought to be time-independent, provided

that the simplified assumptions made in the derivations hold. Figure 5 shows the time series of the global fit coefficients for the exponential and the 3rd order polynomial fit. Coefficients vary more strongly from years 1965 to about 1990 for the



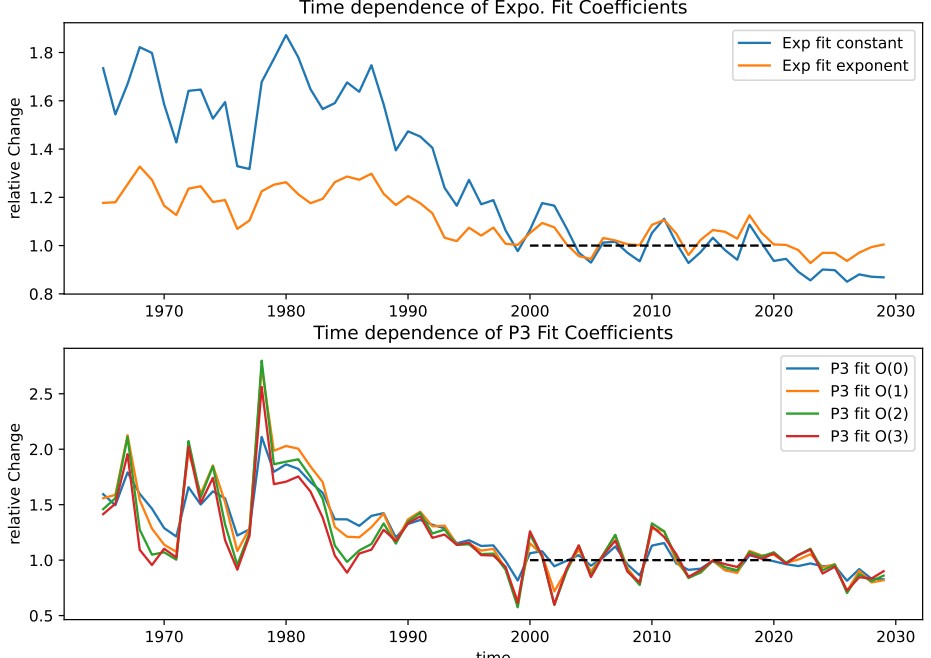

**Figure 5.** Time-dependence of fit coefficients for exponential fit to effective lifetimes $\tau_{eff}$ (top), and with the 3rd order polynomial fit (bottom). All fit coefficients are normalized to the average over years 2000-2019 (period indicated by dashed line).

polynomial and to about year 2000 for the exponential fit coefficients. After year 2000, the fit coefficients show lower variability (with variations mostly smaller than 20%). We interpret the general decline in the fit coefficients as spin-up effect, following the simulation start in 1950. While we estimate that ideal AoA is spun-up sufficiently after about 15 years (i.e., in 1965), the

effective lifetime is more sensitive to the long tail of the spectrum (contributing depleted air to the spectrum), and therefore takes longer to equilibrate. This is confirmed by a similar analysis with a linear increasing $SF_6$ tracer, for which the effective lifetime decreases strongly over the first 20 years, and converges to within 10% of its equilibrium value by year 2000 (not shown). Note that this spin-up effect of the $SF_6$ effective lifetimes are not only model internal, but due to the start of emissions in the 1950s, a similar effect is likely present in the atmosphere. In the following we use mean values of the fit coefficients over

years 2000-2019 (period indicated by dashed line in Fig. 5), for which the values have converged rather well. The deviations in effective lifetime towards the beginning of the simulation might induce errors in the correction, but it is shown in the following section that those are generally small compared to the overall beneficial effect of the sink correction.

## 4.2 Application of correction schemes to model data

As a next step, we apply the correction scheme with different fits to model data to quantify how well the correction scheme is

able to remove biases due to the chemical sinks in $SF_6$-based AoA. In the self-consistent model context where we have the full



information of ideal and apparent AoA we can test whether the assumptions and fits made in the formulation of the correction scheme hold. In particular, we test how well different fit functions perform, and how much the correction improves by using latitudinally-dependent fit coefficients instead of globally averaged ones.

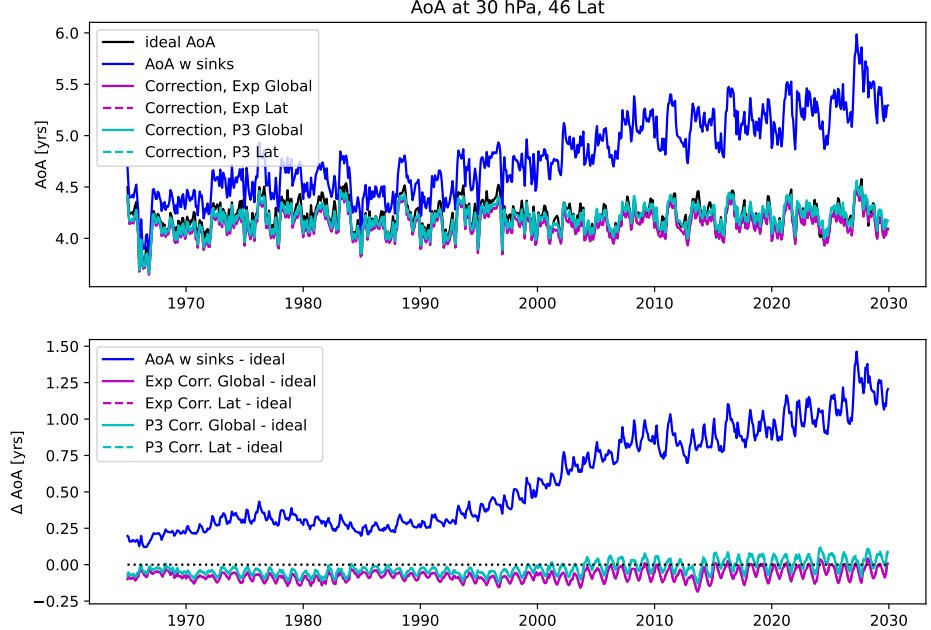

**Figure 6.** Top: Timeseries of ideal mean AoA (black) in mid-latitudes (46°N) and 30 hPa, together with apparent $SF_6$-based AoA (blue), and apparent AoA corrected with the exponential correction scheme (magenta) and the P3 correction scheme (cyan) Bottom: as above, but differences to the ideal mean AoA.

We first present an example time-series at mid-latitudes in the middle stratosphere (at 30 hPa and 46°N, see Fig. 6). Con-

sistent with Loeffel et al. (2022), $SF_6$-derived apparent AoA increases over time, while the ideal AoA remains constant, as expected in the time-slice simulation with constant climate. The difference of apparent AoA to ideal AoA is around 0.5 years in year 2000, and up to 1 year towards the end of the simulation in 2030 (see Fig. 6). Corrections with exponential and 3rd order polynomial fits are performed, both with either latitudinal dependent or global fit coefficients. All four methods are able to strongly reduce the apparent to ideal AoA difference over the entire simulation period to less than about 0.15 years. All cor-

rection schemes over-correct slightly before 2000, likely because of the change in effective life-time over time in this period due to spin-up effects (see Fig. 5). Furthermore, the remaining difference of the corrected AoA to ideal AoA has a seasonal cycle, indicative of seasonal variations in the effective lifetime which are not captured by the annual mean fit coefficients. The correction using an exponential fit performs slightly worse than the correction using the 3rd order polynomial, but differences between the methods are small. There is almost no difference between the corrected AoA using global versus latitudinal

dependent fit coefficients at this location in the mid-latitude middle stratosphere.





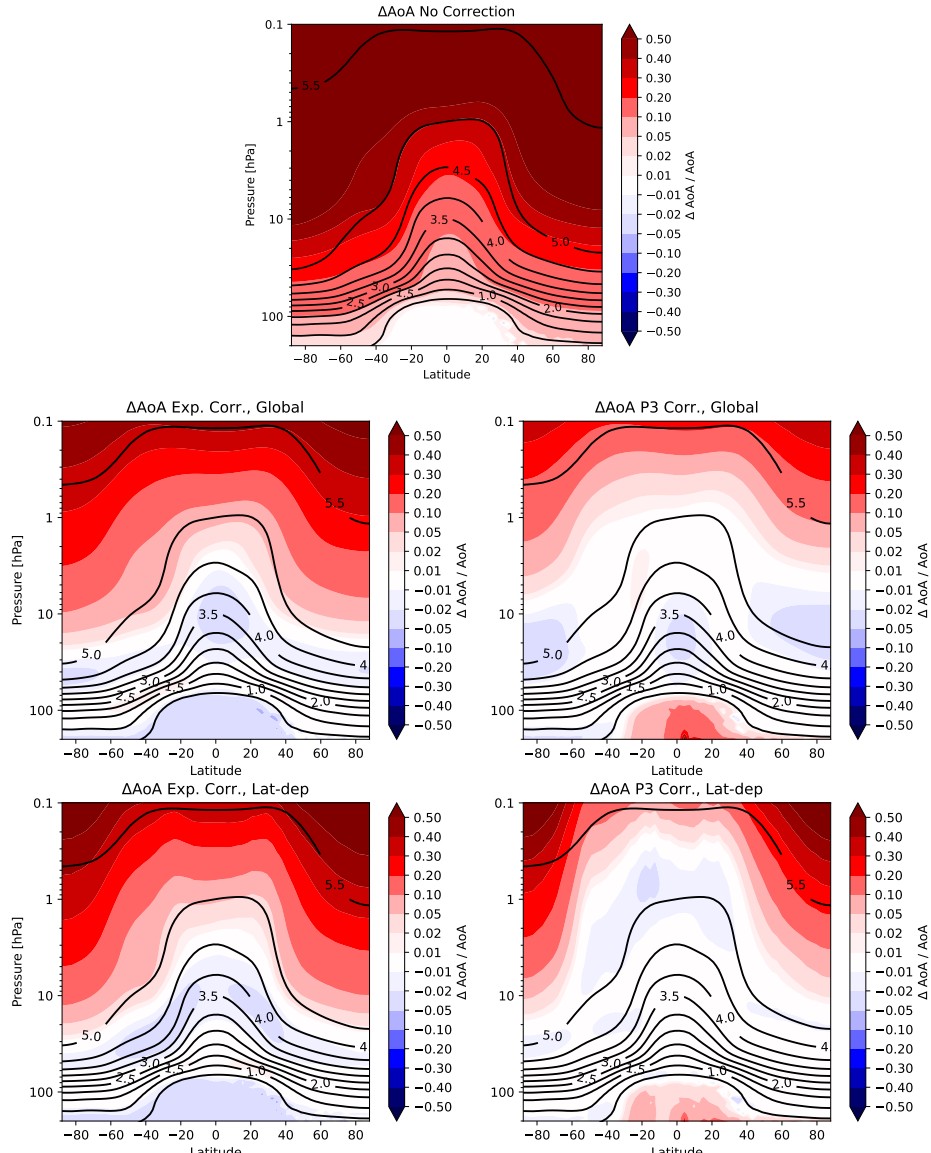

**Figure 7.** Top: Relative difference of apparent mean AoA to ideal mean AoA averaged over 2000-2009 without correction. Black contours show the climatology of ideal AoA in all panels. Middle and bottom panels: Relative difference of apparent to ideal mean AoA as above, but with the sink correction applied using exponential (left) and 3rd order polynomial (right) fits with either global mean fit coefficients (middle row ) and latitudinal-dependent fit coefficients (bottom row). Note the non-linear color bar.

The gain of applying the correction at each latitude and pressure level is demonstrated in Fig. 7 for corrections with the exponential and the 3rd order polynomial, each of them for global and latitude-dependent fit coefficients. For reference, the top panel in Fig. 7 shows the relative difference of apparent to ideal AoA for the uncorrected $SF_6$-derived AoA, showing average





differences of 2-5% for young air of 1-2 years, differences between 5 to 10 % for AoA between 2 to 4 years, and strongly rising

differences for older air. All correction methods are capable of reducing the relative difference below 2% for AoA younger than 4 years in almost all regions of the stratosphere. The polynomial fit even reduces the relative differences to less than 1% in almost all regions up to 5 years of ideal AoA, in particular when using latitudinal dependent fit coefficients. Above ideal AoA of 5 years, the bias in apparent AoA increases strongly. As can be seen in Fig. 3 (right), both the corrections based on the exponential and the linearly extrapolated polynomial fit (see discussion in Sec. 4.1) are for most seasons and latitude bands too

conservative (i.e., undercorrect). Since the fits are performed only for ideal AoA below 5 years, the correction scheme is not designed to remove this bias for very old air completely.

      The reduction in biases in apparent AoA through the sink correction is further quantified in Fig. 8 for different AoA bins. The probability distributions of relative differences are based on monthly, zonal mean data for the decade 2000-2009. The relative differences between apparent corrected and ideal AoA are strongly reduced compared to the uncorrected $SF_6$-derived AoA.

This is true for all correction schemes. The improvement is smallest for AoA below 1 year, where effects of sinks are small (median deviation of less than 5%) and for which the correction scheme is not designed. Nevertheless, applying the correction scheme does not degrade AoA for those young air masses.

      All correction methods reduce the relative differences to below 1% in the median for AoA between 1 and 3 years, and to below 2% for AoA up to 5 years. This corresponds to a factor of 10 or more compared to the median of relative differences

for the uncorrected AoA. The polynomial fit methods perform slightly better than the exponential fit method in particular for AoA between 3 and 5 years, as expected from the better ability to fit the effective lifetimes (see Sec. 4.1). There is little gain when moving from a 3rd order to the 5th order polynomial fit. For AoA outside the 1-5 year range, the 5th order polynomial performed even poorer, likely because outside the fit region higher order polynomials are poorly constrained. Thus, based on those results we conclude that a 3rd order polynomial fit performs overall the best.

Using latitudinal dependent fit coefficients versus global fit coefficients leads to slightly larger reductions of the relative AoA differences in particular for AoA of 2-3 years. The better performance of latitudinal fit coefficients stems mostly from tropical latitudes (see Fig. A1), where the effective lifetimes differ from the remaining latitudes in partiualr for AoA of 2-4 years (see Fig. 4).

      For AoA beyond 5 years, relative differences between uncorrected apparent and ideal AoA become very large, exceeding

50% in the median and reaching up to almost 250% for extreme cases (note the scaled values for this AoA bin in Fig. 8). While the correction is not designed for this AoA range, it is still capable of reducing the relative difference to ideal AoA substantially. In particular the polynomial fit reduces the median of relative differences to below 10%. The latitudinal dependent fit coefficients perform worse than the global fits for those old air masses, stemming from an undercorrection at high latitudes (see Fig. A1 and Fig. 7).

The fit coefficients show substantial changes for earlier time periods (see Fig. 5), but we use the average fit coefficients for 2000-2019 for all times. Despite this, the relative differences between $SF_6$-derived and ideal AoA can also be reduced in the 1980s to below 2% in the median for all ideal AoA ranges between 1 and 5 years, and with all correction methods (not shown).




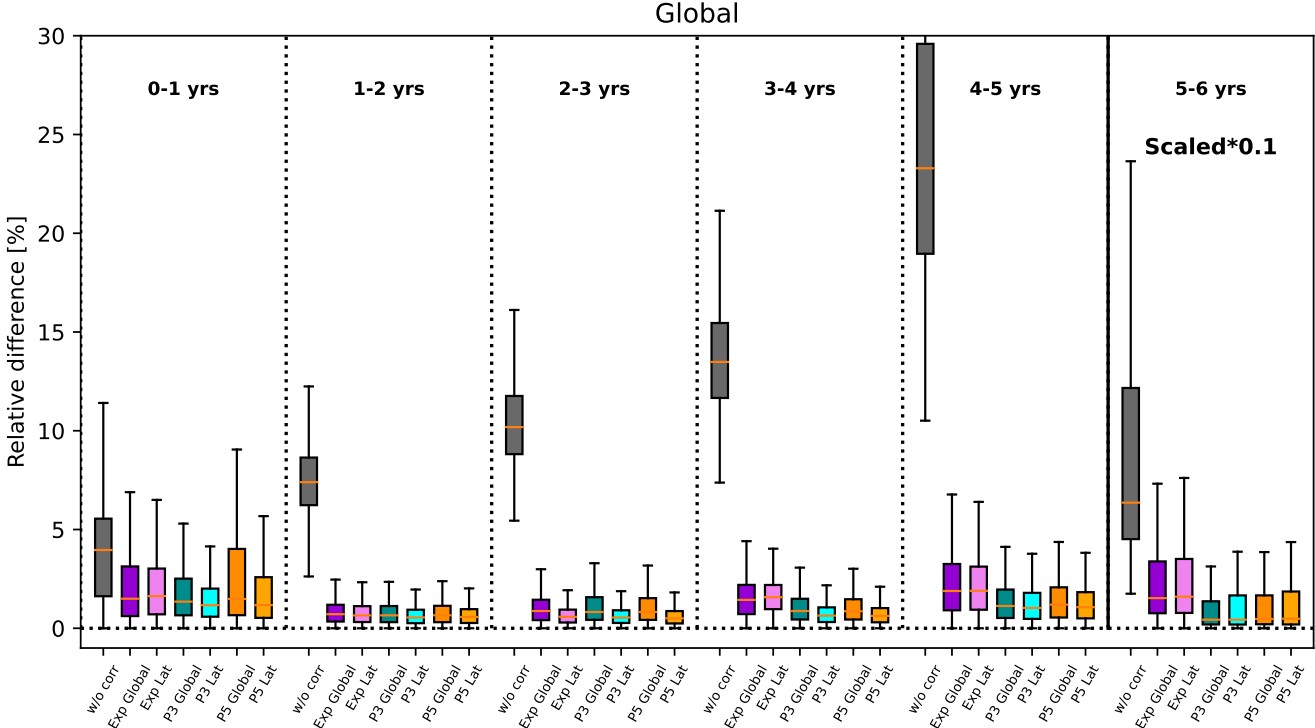

**Figure 8.** Relative differences between ideal AoA and apparent AoA without and with different correction schemes (see x-axis labels) sorted by AoA for years 2000-2009. Note that the values for AoA between 5-6 years are scaled by a factor of 0.1. For the first AoA bin of 0-1 year, AoA below 0.2 years is excluded to avoid excessive relative differences. The coloured boxes show the range of the lower to upper quartile of the data, the orange line is the median and the whiskers indicate the minimum and maximum values.

However, the gain compared to uncorrected AoA is generally smaller, since the effects of sinks on AoA are weaker in the 1980s (ranging from 2% to 9% for AoA between 1 and 5 years).

The increasing effect of $SF_6$ sinks over time induces apparent positive trends in AoA (see Loeffel et al., 2022,  and and Fig. 9b). Since we use a time-slice simulation, trends in ideal AoA are not significant (except a few locations, highlighting the role of natural variability). For the corrected $SF_6$-based AoA, trends are strongly reduced compared to the uncorrected apparent AoA trends and are mostly insignificant for AoA values below about 4 years. However, the correction scheme does not completely remove apparent trends - in particular in mid-latitudes in the lower stratosphere a weak significant trend remains.

Also for AoA values above about 5 years, a significant positive trend remains, in particular for the exponential scheme. This is consistent with the AoA correction not being able to reduce the very strong biases for those very old air masses (see Fig. 7).

Furthermore, we investigate the effects of the sink correction scheme on the difference of tropical to mid-latitude AoA profiles. This is a useful diagnostic, as it can be linked to the strength of the overturning circulation (e.g. Linz et al., 2017). The latitudinal age difference for years 2000-2009 is displayed in Fig. 10 and ideal age shows the typical profile with largest




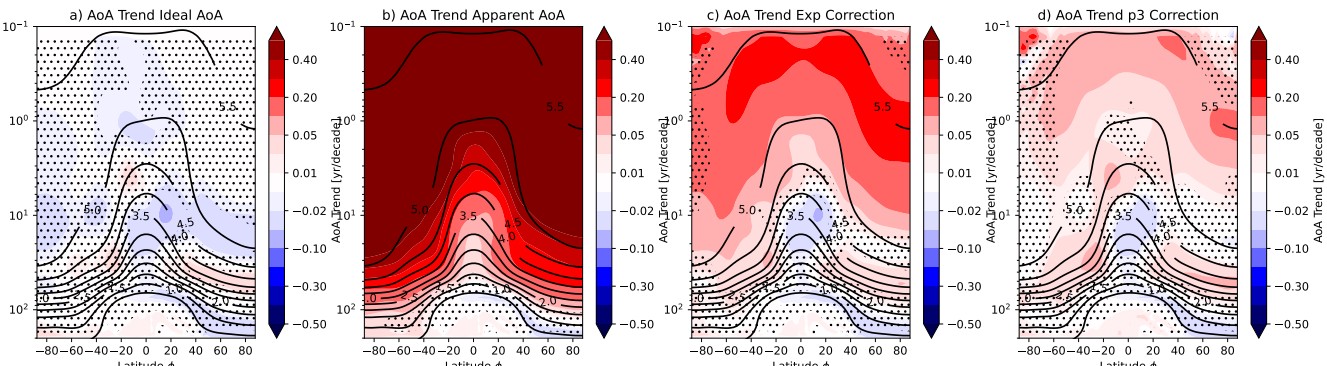

**Figure 9.** Linear Trends in AoA over 1990-2020 in a) ideal AoA , b) apparent AoA and corrected AoA with the c) exponential and d) P3 scheme. Hatching indicates where trends are not significant on a 95% level. Black contours display the ideal AoA climatology.

latitudinal differences in ideal AoA in the lower stratosphere around 60 hPa (e.g. Dietmüller et al., 2018). Apparent AoA strongly overestimates the AoA difference (by about 30% at 60 hPa, increasing to more than 100% above 10 hPa). The corrected AoA is able to remove this overestimation of the AoA difference. In particular with the P3 correction the AoA difference is reproduced remarkably well up to about 3 hPa. Interestingly, the latitudinal dependent fit coefficients are not beneficial for the AoA difference diagnostic, but perform as well as the correction schemes with global mean coefficients.

## 4.3 Sub-sampled data

In the previous Sections, we examined the skill of the sink correction scheme when all model data is available for estimating the fit coefficients. However, the goal is to apply the sink correction to observational data and ideally constraint the coefficients for the correction based on observations. However, from observations we have much less information on the "ideal" to apparent AoA relation. Therefore, we test here by how much the fits and the correction degrades if only limited and uncertain data pairs of ideal and apparent AoA are available to perform the fits on.

To do so, we select data from the model simulation representative of a limited number of observational data sets, and choose data from one selected month and the latitude and height region of observations for each of the samples. We choose here 10 different data samples, representative of the observational data we have available for comparison (see Sec. 3.2 for a description of the data sets). The subsampled data are almost all located in the northern hemisphere, mostly in mid-latitudes with a few tropical and high-latitude profiles. Furthermore, only four profiles reach to or just above 30 km height, i.e., older air is underrepresented. The apparent to ideal AoA differences for the subsampled data are shown in Fig. 11 (left). Due to the limited data, only one global fit is performed, since the data is not sufficient for latitudinal dependent fits. However, as shown in the last Section, using global instead of latitudinal dependent fit coefficients does not degrade the correction significantly.

We perform a bootstrap procedure to estimate the uncertainty in the fit coefficients. Specifically, we choose 141 samples from the 141 available data points with replacement, and to each apparent and ideal AoA, a random error drawn from a normal



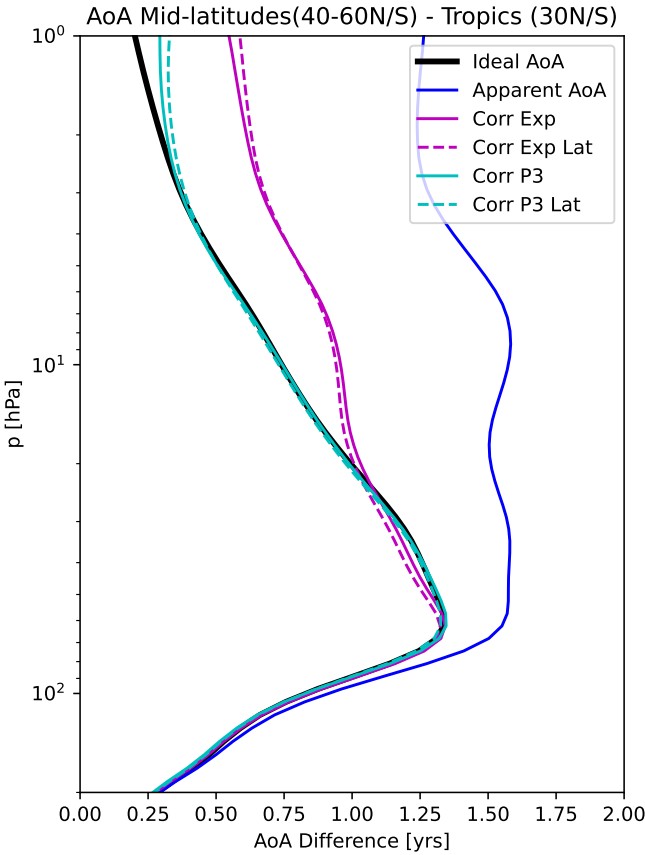

**Figure 10.** Mid-latitude (40-60 N/S) to tropical (30N-30S) AoA differences averaged over years 2000-2009 for ideal (black), apparent (blue) and corrected AoA (cyan and magenta for 3rd order polynomial and exponential correction).

distribution with standard deviation of 0.1 years is added. This error is roughly representative of the uncertainty in observational data (see Sec. 3.2). We repeat this procedure 1000 times, and calculate the average and standard deviation of the fit coefficients from this suite.

As seen in Fig. 11, the resulting fit line for the exponential fit to the sub-sampled data (magenta) lies below the fit based on the full model data (black). The fit varies moderately within two standard deviations, resulting in a correction at 5 year AoA spanning between about 0.65 to 0.95 years. The 3rd order polynomial fit, on the other hand, is very unconstrained for the subsampled data (blue dotted lines). The two standard deviation range spans essentially the whole domain; therefore only few example fits are shown to demonstrate the unconstrained fit.

The effects of the uncertainty in the fit coefficients on the correction of AoA are summarized in the error statistics in Fig. 12. The correction was performed for fit coefficients plus and minus two standard deviations from its best estimate for the fits to the subsampled data. The probability distributions show relative differences including the corrections with both the fit





coefficients plus and minus two standard deviations. The relative differences between ideal AoA and corrected AoA with the 3rd order polynomial fit are very large and span a wide range, consistent with the unconstrained fit lines found above. Thus, we conclude that correction with a 3rd order polynomial fit based on a limited amount and/or uncertain data is not advisable, as

the coefficients are not well constrained. The exponential fit, on the other hand, is much more robust. The relative difference of corrected to ideal AoA is larger when using the exponential fit coefficients obtained from the sub-sampled data compared to the full model data for all AoA bins. Nevertheless, the correction still reduces the error compared to uncorrected data substantially (to below 5% for AoA between 1-4 years, and below 10% for AoA between 4-5 years). In conclusion, despite the fact that limited information of the relation of $SF_6$-derived and ideal AoA leads to degraded fits, the correction scheme based on the

global exponential fit is still very beneficial compared to using uncorrected $SF_6$-based AoA.

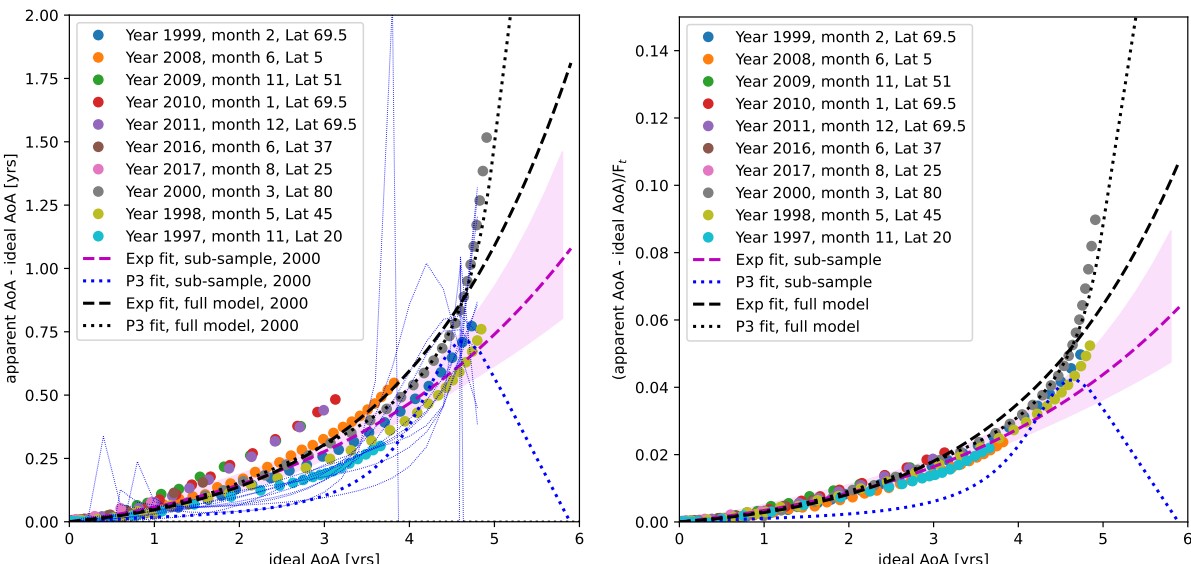

**Figure 11.** Left: Difference of apparent AoA to ideal AoA from model AoA data subsampled to reduced data set representative of observational data availability. Shown are fit lines for full model data (black) and for subsampled data (magenta for expoential fit, blue for 3rd order polynomial fit). The violet shading indicated the uncertainty in the exponential fit to the subsampled data estimated via bootstrapping (see text). Uncertainty from 3rd order polynomial fit is omitted as it spans the whole domain; instead, 10 randomly chosen example fit lines are shown (blue, thin lines). Right: as left, but y-axis normalized by $F_t$ to remove the time-dependence of the sink effect. Individual fit lines for polynomial fit are omitted in this panel for clarity.

## 5 Comparison and application to observational data

In the last section, the performance of the correction scheme was carefully evaluated based on model data, and it was found that the schemes are highly beneficial to remove biases in $SF_6$ based AoA estimates. However, this was entirely based on model data, and one can question whether the chemical sink of $SF_6$ is well enough constrained in the model to trust the



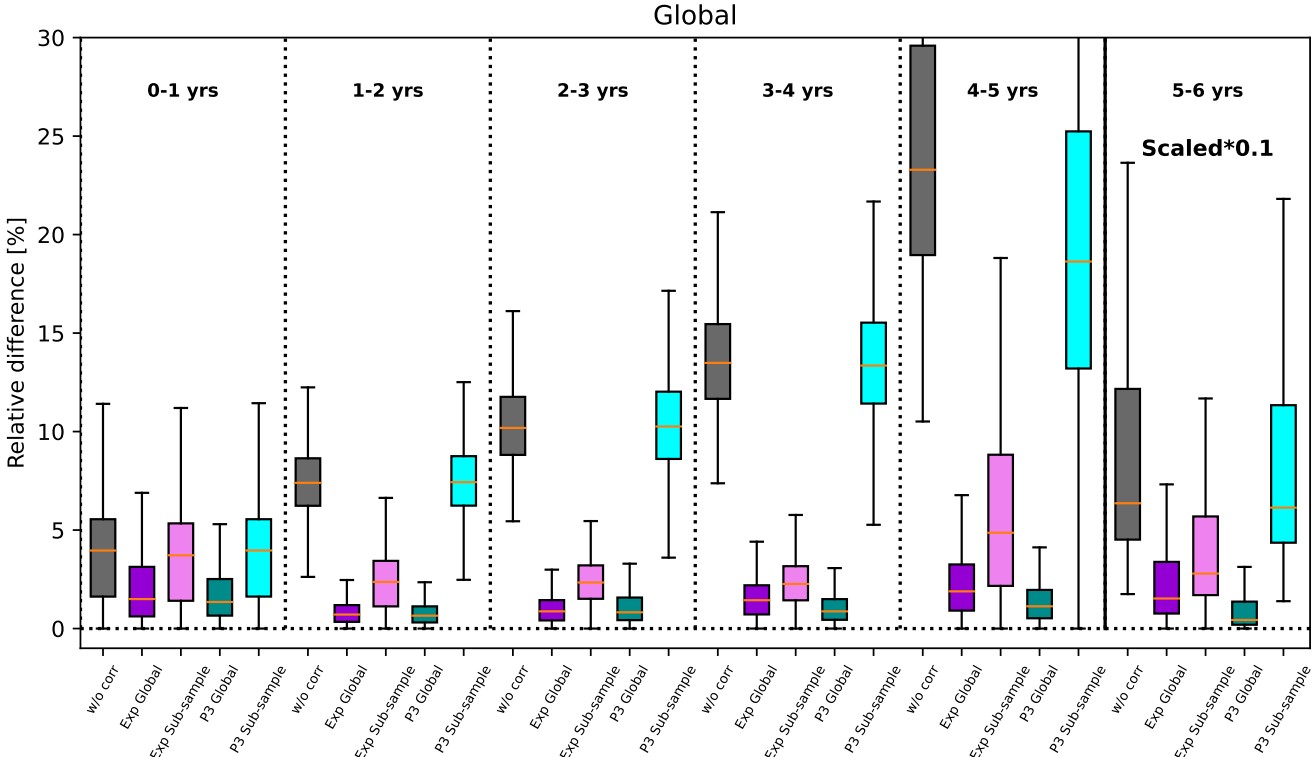

**Figure 12.** As Fig. 8, but contrasting correction with sub-sampled to full model data fit coefficients. The errors in correction with fits to sub-sampled data include fits with the average fit coefficients +/- 2 standard deviations.

derived coefficients. Ideally, we would perform similar fits between AoA derived from observed $SF_6$ and from simultaneous observations of other age tracers, that do not suffer from the sinks. However, as we will show in the following, available observational data is currently not able to constrain the fit parameters well enough. Instead, we will show that the model-derived correction fit lines lie within the bounds given by observations, and we will test the performance of the correction with model-based parameters on independent AoA observations.

We use the observational data sets described in Sec. 3.2 from a number of aircraft campaigns and balloon flights. Figure 13 (left) shows differences between $SF_6$-based AoA and AoA based on the alternative tracers as a function of the latter, similar to Fig. 4 (left) but for observational data. It is immediately clear from this figure that observations show a large spread in the difference between $SF_6$ and alternative tracer based AoA. In the left panel, the difference is compared to model data sampled in the same month and years as the observational campaigns took place. The model data are still monthly and zonal mean profiles, explaining the much reduced variability compared to observational data. Furthermore, note that the meteorological conditions in the free-running model of course do not match the observed state of the atmosphere, so the sampling is rather meant to capture variations due to the latitude, season and the underlying trend in $SF_6$. Generally, the model data is found to



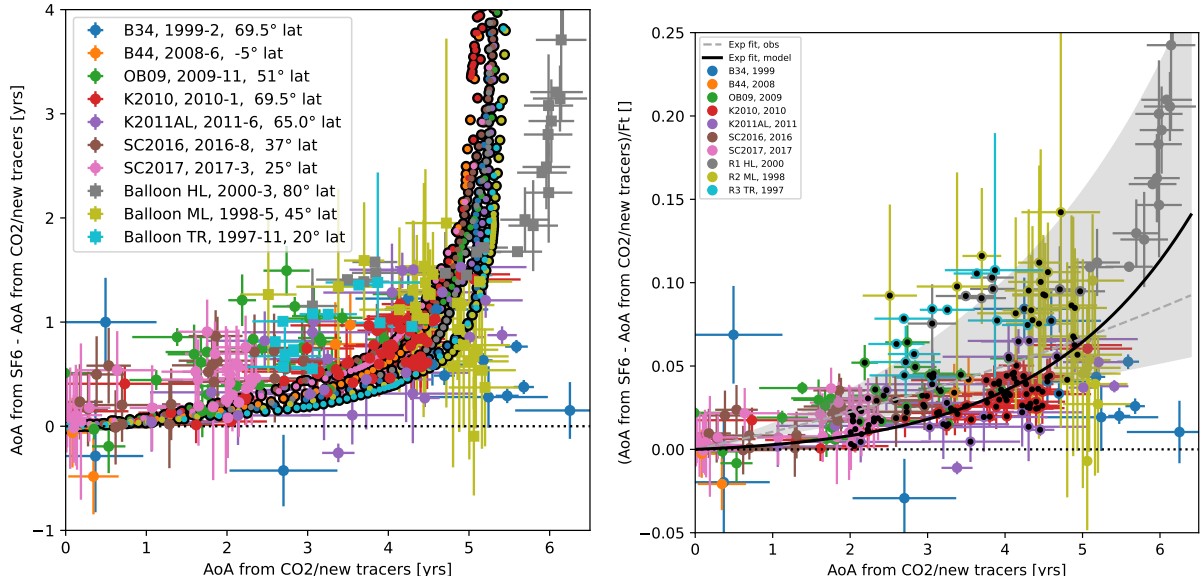

**Figure 13.** Left: Difference of apparent $SF_6$-based AoA to AoA derived from alternative age tracers for a number of aircraft and balloon measurements (see legend). Monthly and zonal mean model data for the same latitude bands and months are shown in same colour, but with black circle. Right: as left, but y-axis normalized by the function $F_t(t)$ to adjust for different years of observations. Exponential fit based on all model data is shown in black, and fit based on chosen observations (indicated with black dots) in grey, with uncertainty as grey shading.

lie within the large scatter of observational data for mean AoA below about 5 years. One high-latitude balloon flight sampled air within the polar vortex (Ray et al., 2017) with $CO_2$-based mean AoA between 5 to 6 years. For this profile, the difference

in $CO_2$ and $SF_6$-based AoA increases strongly to up to 4 years. The AoA differences in the model from a similar latitude band and month as this balloon flight show a similarly strong increase, but occurring already at younger ideal mean AoA values.

The measurement campaigns took place during different years within a 21 year period (between 1997 to 2017), so that the effect of the sinks increased over this time frame. To eliminate this time-dependence, the age differences are normalized by the time-correction function $F_t$ (see Section 2) and the resulting data points are shown in the right panel of Fig. 13. The scatter is

somewhat reduced compared to the uncorrected differences on the left, but is still substantial. Attempts to fit the parameters for the correction scheme result in a very large uncertainty. In particular for young AoA values below 2 years, some measurements indicate already a large difference to $SF_6$-based AoA, ranging between 0 to 1 year (see Fig. 13 left). This might partly be due to local influence close to sources of the alternative AoA tracers, potentially leading to biases in the estimate of both "ideal" AoA from alternative tracers and/or $SF_6$-based AoA (Adcock et al., 2021). In the last section, we found that the model

sub-sampled data to the availability of observational data are not able to constrain the 3rd order polynomial fit. Therefore we only use the exponential fit method in the following, both for attempts to fit coefficients based on observations and for any corrections of the observational data. When discarding all mean AoA values below 2 years and above 5 years, the exponential fit for the correction scheme as displayed in grey in Fig. 13 (right) is obtained (based on a bootstrapping algorithm to estimate




the uncertainty, see grey shading). However, the fit is very sensitive to in- or excluding individual data points. When including

data between 1 to 2 years, the resulting fit would indicate an increase in the effective lifetime with mean AoA, which can

be ruled out as nonphysical. Therefore, we have to conclude that the currently available observational data that allow for an

estimation of $SF_6$-based AoA and AoA from an alternative tracer are not sufficient in quantity and quality to constraint the fit

parameters for the sink correction.

Instead, we have to rely on the model-based parameters for the correction scheme. The model-based global mean correction

curve for the exponential fit is included in Fig. 13 (right; black curve). While the large scatter of observations do not allow

for a detailed evaluation, we find that the model-based correction scheme lies within the observational uncertainty. Therefore,

we suggest at this stage as best practice to apply the sink correction with the model-based parameters. While higher order

fit functions than the exponential fit, as well as latitudinal dependent fit coefficient did lead to a slightly better performance

of the correction within the model (see Sec. 4.2), we rather use the more conservative exponential fit and the global mean

fit parameters in the following to correct observational data. This is on the one hand because of the larger sensitivity in the

polynomial fits (see Sec. 4.3), and on the other hand because we want to reduce model-depended information. In particular the

latitude-dependence of fit coefficients likely are a function of the representation of circulation in the model (e.g., location and

strength of the polar vortex), while we assume the global fit to be less sensitive to the representation of circulation in a particular

model. Comparisons to other independent observational data and inter-comparisons with other models with implementation of

chemical sinks of $SF_6$ should be performed in the future to better constrain the fit coefficients of the sink correction.

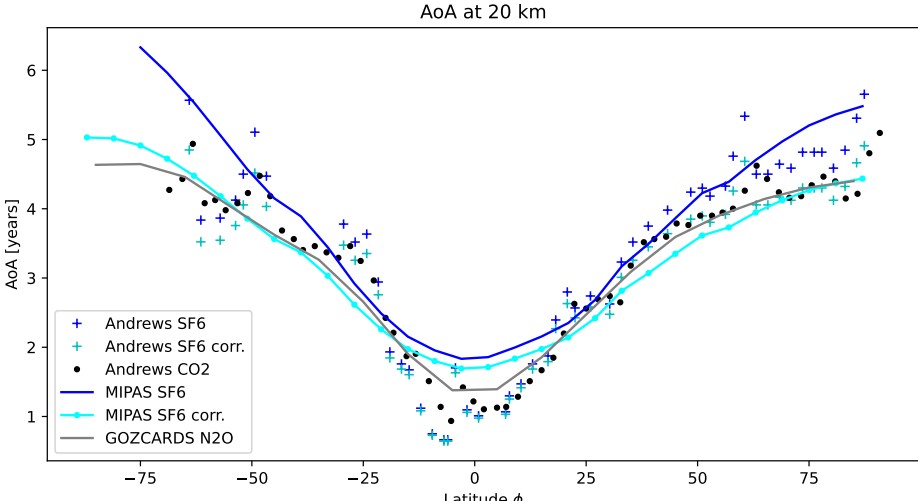

**Figure 14.** Observational mean AoA data at 20 km altitude from in-situ observations by Andrews et al. (2001) based on $CO_2$ (black dots) and $SF_6$, original estimates (dark blue crosses) and with the sink correction applies (lightblue crosses). Mean AoA from MIPAS derived from $SF_6$ (Stiller et al., 2021) and averaged over the whole data record is shown in dark blue solid line and the sink corrected values as lightblue solid line. Mean AoA derived from GOZCARDS $N_2O$ (Linz et al., 2017) and averaged over the entire record is shown as gray solid line.





The correction scheme with the model-derived parameters is applied to in-situ and satellite data, presented in Fig. 14. The latitudinal profiles of in-situ mean AoA data compiled by Andrews et al. (2001) at around 20 km consists of both AoA based on $CO_2$ and based on $SF_6$. The $SF_6$-based mean AoA data consistently lie above AoA based on $CO_2$, in particular for mid- to high latitudes. This difference is not necessarily only due to the chemical $SF_6$ sink, but could also be caused by the different temporal slopes and seasonality of the tracers time-series, affecting the derivation of mean AoA (see e.g. Andrews et al., 2001; Bönisch et al., 2009).

When applying the sink correction to the $SF_6$-based mean AoA values, this consistent high bias of $SF_6$-based mean AoA is strongly reduced. In particular at northern mid- to high latitudes, $CO_2$ and $SF_6$-based AoA agree extremely well after the sink correction is applied, with the mean difference of $CO_2$ to $SF_6$ AoA for latitudes > 40° being reduced from 0.43 years to 0.03 years. In the southern hemisphere, the scatter is larger due to generally poorer data quality. Nevertheless, the bias between $CO_2$ and $SF_6$-based AoA is also reduced from 0.33 to 0.13 years for latitudes south of 40°S.

Furthermore the latitudinal climatological profile of AoA around 20 km is shown for two satellite products, $SF_6$-based mean AoA from MIPAS (Stiller et al., 2021, and see Sec. 3.2) and AoA derived from $N_2O$ measured by MLS and ACE (the GOZCARDS data product, see Sec. 3.2). The $SF_6$-based mean AoA from MIPAS is older in particular at high latitudes compared to AoA based on $N_2O$ and in-situ $CO_2$ observations, but agrees overall well with the in-situ $SF_6$-based mean AoA. After applying the sink correction to MIPAS AoA data, agreement between MIPAS AoA and the non $SF_6$-based AoA data sets is much improved in particular at high latitudes. At mid-latitudes, corrected MIPAS mean AoA is slightly lower than the other estimates, while in the tropics the high bias is only slightly reduced.

Overall, mean AoA estimates from different trace gases and from different observational sources agree much better after applying the $SF_6$ sink correction. This provides further evidence that the sink correction is a promising way forward to remove biases in mean AoA estimated based on $SF_6$.

## 6 Summary and Conclusion

In this paper, we have developed a correction scheme for $SF_6$-based AoA to remove the biases generated by chemical $SF_6$ sinks. The correction scheme is based on a simplified formulation derived from theory, that provides a relation of ideal to apparent AoA dependent on 1) a time-dependent function of the reference $SF_6$ surface mixing ratio (which would be a linear function for a linear reference time-series), and on 2) the path-integrated average effective lifetime of AoA. For the former, we suggest a simplified approximation based on the linearized reference time-series. For the latter, we parameterize the effective lifetime as a function of ideal AoA. Based on global model data, we find that either an exponential or a 3rd order polynomial fit is suited for this parameterization. While the 3rd order polynomial fit captures the details of the AoA-dependency of the effective lifetime better, it comes at the cost of high parameter sensitivity to uncertainty in the underlying data. Furthermore, we find that using latitudinal-dependent coefficients for the parameterization of the effective lifetime as a function of ideal AoA is beneficial, but the overall ability to reduce biases in $SF_6$-based AoA is only marginally improved. In the model, the simplified sink correction with global mean fit coefficients is able to reduce the biases in AoA by a factor of 5 to 10 to less than



2% relative to ideal AoA in the mean for AoA below 5 years. For AoA above 5 years, the relation between ideal and apparent
AoA becomes weaker, and therefore the correction is not designed for those high ages above 5 years. Nevertheless, correcting
AoA with the extrapolated fit lines for this AoA range also substantially reduced the bias. In addition to mean AoA, we showed
that the correction scheme is able to remove apparent AoA trends in the lower stratosphere. Furthermore, the sink correction is
able to correct the tropics to mid-latitude age difference, a diagnostic of the residual circulation strength, even when based on
global mean fit coefficients.

In conclusion, the sink correction scheme performs very well within the self-consistent "model world" with all information
available. However, how useful is this scheme to correct observational AoA estimates given imperfect knowledge of the ideal
to apparent AoA relationship? Firstly, we approach this question by sub-sampling the model data leading to erroneous fit
coefficients. When using the less sensitive exponential fit function for the relation of the effective lifetime to AoA, we could
show that despite incomplete knowledge and thus uncertain fit coefficients, the sink correction is able to reduce the AoA
biases in all cases. In other words, even applying the sink correction with uncertain parameters is better than not applying
any correction. Of course this statement hinges on staying within a certain range of fit uncertainty; if the sink in the model
was completely different than the real world this would naturally not hold. The total stratospheric lifetime of $SF_6$ estimated
in the EMAC model simulations of 2100 years falls within the range of values provided by previous model and observational
studies (e.g. Ray et al., 2017; Kovács et al., 2017; Kouznetsov et al., 2020, providing ranges between around 800 to 3000
years), indicating the validity of EMAC's $SF_6$ depletion mechanisms despite existing uncertainties in lifetime determination
(see also discussion in  Loeffel et al., 2022). Further, comparisons of the model fit to available observational data of $SF_6$-based
AoA and AoA deduced from $CO_2$ or the new alternative AoA tracers (Leedham Elvidge et al., 2018) indicate that the model
sinks do lie within the observational spread. The spread is however high, likely because of two reasons: Firstly, AoA deduced
from realistic tracers is generally error-prone due to uncertainty in the reference time-series, non-linearity therein (including the
seasonal cycle, in particular for $CO_2$) and e.g., influence of local sources of the tracers. Those factors induce the large error bars
in observational AoA (seen in Fig. 13). Secondly, the observational data used here are point measurements, while in the model
we used monthly mean, zonal mean data. While the averaged data fall on the compact ideal AoA to apparent AoA relationship,
point data might differ from this line due to local mixing between different air masses Plumb (as is well known for tracer-tracer
relationships, see e.g. 2002). It is not clear how strong the contribution of those two factors is to the large spread in the observed
relationship of $SF_6$-based AoA to AoA from other tracers. If the contribution from internal variability by mixing is substantial,
this would imply that the compactness of the $SF_6$-AoA to ideal AoA relationship is limited for point measurements, and thus
a correction is not easily possible. Until this issue is clarified, we recommend to use the sink correction scheme on averaged
data (over time periods of a month or so and/or large regions). Application of the sink correction to satellite and compiled
in-situ data confirm that the correction is able to reduce the discrepancy between different AoA estimates. In-situ $SF_6$ and
$CO_2$-based AoA agree very well after the correction is applied, and also $SF_6$-based AoA deduced from MIPAS satellite
observations agrees well to $N_2O$-based AoA from the merged GOZCARDS satellite product when the correction is applied.
Note, however that AoA derived from $N_2O$ also bears uncertainties: AoA is calculated from the $N_2O$-AoA relationship based
on specific concurrent observations of $N_2O$ and $CO_2$ in the northern mid-latitudes (see Sec. 3.2), and improved quantification



of the latitudinal and temporal variations of the $N_2O$-AoA relation will be necessary to more reliable derive mean AoA by this
method.

Overall, we conclude that observational data is currently not able to constrain the parameters for the sink correction scheme
proposed here. Until this is possible, we recommend to use the model-based global mean exponential fit parameters for the sink
correction. While the model might be subject to biases, we could demonstrate that the ideal to apparent AoA relation from the
model lies within observational uncertainty, and that application of the model-based sink correction is successfully reducing
differences in observational AoA estimates. However, future work is needed to test and better constraint the parameters for the
sink correction scheme.

*Code and data availability.* The Modular Earth Submodel System (MESSy) is continuously further developed and applied by a consortium
of institutions. The usage of MESSy and access to the source code is licenced to all affiliates of institutions which are members of the MESSy
Consortium. Institutions can become a member of the MESSy Consortium by signing the MESSy Memorandum of Understanding. More
information can be found on the MESSy Consortium Website (http://www.messy-interface.org). The simulation presented here has been
carried out with MESSy version 2.54.0. Python code to perform the sink correction is provided as supplmenet to this paper. The EMAC
simulation data are archived at DKRZ and are available upon request to the authors, observational data are available upon request.

**Appendix A: Error statistics for tropics and high latitudes.**

In addition to the global statistics of the effects of the sink correction on the apparent - ideal AoA difference, in Fig. A1 those
error statistics are presented in the same manner for AoA only in the tropics (20S-20N, top) and for southern high latitudes
(60-90S, bottom panel).

*Author contributions.* HG designed the study, performed the analysis and wrote the manuscript. RE performed the model simulations. JL,
ER, GS, HB and LS provided observational data, and contributed to the design of the sink correction. All authors contributed to improve the
manuscript.

*Competing interests.* At least one of the (co-)authors is a member of the editorial board of Atmospheric Chemistry and Physics.

*Acknowledgements.* This paper arises as outcome from the International Space Science Institute (ISSI) project on "Stratospheric Age-of-Air:
Reconciling Observations and Models". We thank ISSI for support of two team meetings held in Bern. The authors thank all team members
for stimulating discussion and comments on the work. We thank Matthias Nuetzel for comments on the manuscript. RE acknowledges support
by the Czech Science Foundation under Grant No. 21-03295S.







**Figure A1.** As Fig. 8 but for individual regions: Tropics (20S-20N), SH high latitudes (60-90 S). Northern high latitudes show very similar results to the SH (not shown).



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
