# Peer review of "Correction of stratospheric age-of-air derived from $SF_6$ for the effect of chemical sinks"

_EGUsphere, 2023_

## Author Comment (AC1)

**Reply to Reviewer comments on "Correction of stratospheric age-of-air derived from SF6 for the effect of chemical sinks"**

We thank the two reviewers for their evaluation of our manuscript and the constructive comments which we believe helped to improve our study. In particular in response to the major comment by Reviewer 1, we revised the theoretical derivation of the expression the sink correction scheme is based on, realizing the crude single path assumption is not necessary to obtain the expression. Please find below a point-to-point response to all reviewer comments.

**Reviewer 1**

The manuscript proposes a methodology for correcting SF6-based age-of-air (AOA) to remove biases associated with chemical sinks. I commend the authors for tackling this difficult problem, which I feel they address rigorously within the limitations of their approach. However, I have several concerns about certain fundamental assumptions that are made and these concerns subsequently lead me to encourage publication only after major revisions have been performed. In particular, I am concerned about the approximation of the AOA spectrum as a delta function (major comment below), which is clearly violated throughout the entire stratosphere. I ask that the authors provide more solid physical evidence for the assumptions underlying their methodological approach.

**Major Comment:**

I have major concerns regarding the assumption introduced in developing equation (2) that the AOA spectrum (i.e., G) be approximated as a delta-function. Physically, this assumption only approximates the true physical medium when the flow is purely advective (i.e., mixing = 0). This is simply *not* the case in the stratosphere and, indeed, is the whole reason why the age spectrum was developed thoroughly in Hall and Plumb (1994). While the authors seem to appreciate that this is a "crude assumption" I find it too inconsistent with the physical nature of stratospheric transport to be used. I ask that the authors more rigorously justify their assumption and/or modify their derivation to account for the effects of mixing, as this assumption renders the whole methodology seriously limiting in its applicability.

Hall, Timothy M., and R. Alan Plumb. "Age as a diagnostic of stratospheric transport." Journal of Geophysical Research: Atmospheres 99, no. D1 (1994): 1059-1070.

We absolutely agree with the reviewer that a delta-function age spectrum is a bad, non-physical assumption, and we thank the reviewer for the comment as this triggered us to re-assess the necessary assumptions for our theoretical derivations. In this process, we realized that the delta function assumptions is unnecessary, and the same formulation of the relation of the apparent to the ideal age can be obtained without making this crude assumption. We refer the reviewer to the revised Section 2 for details (see paragraphs "Linearly increasing tracer" and "Non-linearly increasing tracer") .

The only assumptions that still need to be made are now clearly stated (see lines 90 ff in revised paper), namely 1) the assumption that the lifetime is large compared to the transit times (which we argue is well justified within the stratosphere), and 2) the assumption that the generally transit-time dependent lifetime can be approximated with a mean path-averaged lifetime (see also comment below).

**Minor Comment 1:**

How does the "path-averaged lifetime" introduced in equation (2) relate to the "path-dependent lifetime" introduced in Holzer and Waugh (2015)? It appears to me that the authors may be acting unconstructively (though, perhaps, unwittingly) here by adding unnecessarily new terminology to the field. Please provide a rigorous definition of this lifetime and relate it to that from this 2015 study. Note that the path-dependent lifetime presents the advantage that it does **not** assume zero mixing.

Holzer, Mark, and Darryn W. Waugh. "Interhemispheric transit time distributions and path-dependent lifetimes constrained by measurements of SF6, CFCs, and CFC replacements." Geophysical Research Letters 42, no. 11 (2015): 4581-4589.

Thank you for encouraging us to relate the "effective lifetime" defined in our study to previous work. You are correct, the "path-dependent lifetime" defined in Holzer and Waugh (2015) is essentially identical to the assumption of using an effective lifetime in our paper. We have added a remark when introducing the effective lifetime, also referencing to the "average path assumption" by Schoeberl et al (2000), which is yet a similar approach (see new lines 92-96):

*"Secondly, we will make the assumption that the transit time dependent lifetime τ (t') can be replaced by the path-average lifetime τ_eff . This lifetime τ_eff is the path-average across all possible paths to the point of interest of the path-integrated lifetime τ (t'), and we refer to this as the "effective" lifetime. Note that similar formulations have been used previously by Holzer and Waugh (2015), where they refer to this quantity as the "path-dependent inverse loss frequency", and by Schoeberl et al. (2000) in their "average path assumption".*

We have nevertheless chosen to keep the term "effective lifetime" for two reasons: 1) later in the paper, we derive and fit the effective lifetime purely as function of mean age, and base its estimation on the approximate expression (Eq. (10) in the revised manuscript); any validations of the assumptions made to derive this expression thus will alter the derived effective lifetime, making "*effective*" a suitable choice. 2) this might be a matter of personal taste, but personally I find the expression "path-dependent" for a quantity that is averaged across and along different paths not entirely intuitive.

**Minor Comment 2:**

Equation (1): If you are defining t' as transit time (somewhat unconventional, as transit time is often denoted using the notation tau=t-t', see Holzer and Hall (2000) for more), then your use of G(t') suggests that the transport operator is stationary, i.e., not dependent on either the initial pulse time or receptor time. This assumption needs to be stated clearly as several studies have shown that this assumption is actually *not* valid, even in the stratosphere. In particular, see the study by Li et al. (2012), which directly quantifies the large non-stationarity of G (and associated mean age):

Holzer, Mark, and Timothy M. Hall. "Transit-time and tracer-age distributions in geophysical flows." Journal of the atmospheric sciences 57, no. 21 (2000): 3539-3558.

Li, Feng, Darryn W. Waugh, Anne R. Douglass, Paul A. Newman, Steven Pawson, Richard S. Stolarski, Susan E. Strahan, and J. Eric Nielsen. "Seasonal variations of stratospheric age spectra in the Goddard Earth Observing System Chemistry Climate Model (GEOSCCM)." Journal of Geophysical Research: Atmospheres 117, no. D5 (2012).

We agree that the Green's function is time- and space dependent, and thank the reviewers for pointing out that we have missed to state this clearly. We have now included the dependency in Eq. (1) as G(t',t,x), and explicitly stated that this dependency has been dropped in the following due to brevity.

**Minor Comment 3:**

Is it really prudent to use the balloon data from 2000 (grey stars on Figure 2)? These large values seem unrealistically high.

The balloon data from 2000 that the reviewer refers to are the only observed profile we use here from within the winter polar vortex. It is well known that SF6-based mean AoA becomes excessively high in the winter polar vortex, because of downward transport of SF6-depleted air (see e.g. Ray et al, 2017; Loeffel et al., 2022 and references therein). Later in the paper, we argue that our approach to correct for the SF6-sinks only works reliably for mean age below ~ 5 years, thus we do not include those data points in any of the fits. The data are useful, though, to give an estimate of whether the (fits from) model data are in the range of the observed values (see e.g. Fig. 11, 13), thus we decided not to omit them.

**Typos:**

Line 36: remove "and" after "surface"

reformulated to " *AoA is simply the lag time between the mixing ratio at the surface and the mixing ratio at a given point in the stratosphere.*"

Line 47: typo "particualr"

thanks, corrected.

Line 56: Do you mean "ideal age", not simply "ideal"? If so, please diligently acknowledge the following references:

Thiele, G., and J. L. Sarmiento. "Tracer dating and ocean ventilation." Journal of Geophysical Research: Oceans 95, no. C6 (1990): 9377-9391.

England, Matthew H. "The age of water and ventilation timescales in a global ocean model." Journal of Physical Oceanography 25, no. 11 (1995): 2756-2777.

yes, we mean "ideal age" and modified the text to clarify.

Thank you for pointing out the references to the age concept used in ocean circulation. We appreciate that there is strong overlap in the concept of using "age" in the stratosphere and the ocean, but given in the particular sentence around line 56, we refer explicitly to effects of chemical sinks on estimations of mean age, we found that the references would not be a good fit for this issue.

Line 61: Again, please clarify the use of "ideal" here.

Done.

**Reviewer 2**

This is a well-written and thorough paper that is well suited for publication in ACP. The derivation of AoA from observed SF6 is an important topic in stratospheric research and this paper is a novel very useful contribution to the challenge of how to best quantify AoA from a tracer with chemical sinks. Overall I think that this paper represents a significant amount of intellectual and modelling work and I only have a few minor comments before I think it is publishable (see below).

There are likely further details and approximations which could be explored by different model setups but I think the current paper is largely sufficient to present the proposed methodology

Specific comments.

The model uses a timeslice setup which will give a steady circulation (with variability). I understand the reasons for that. However, it would be good to mention the possible implications (if any) for the work if a transient run had been used to simulate up to the present day, assuming the model would capture any realistic trend in AoA.

Thank you for the comment, and we agree that it would be interesting to test the proposed correction scheme in a transient simulation. In our preceding study by Loeffel et al (2022; https://acp.copernicus.org/articles/22/1175/2022/acp-22-1175-2022.html), we had compared results from different simulation set-ups, including the timeslice simulations used here as well as a transient simulation. In this study, we had shown that the increase over time in the difference of SF6-based mean age to ideal mean age is identical within error margins in the timeslice simulation and the transient simulation (see Table 3 in Loeffel et al (2022): Trend at northern mid-latitudes at 30 km altitude in transient simulation (REF) over 1965-2011 of tracer with sinks minus tracer without sinks is 0.25 +/- 0.01 year/decade, and in TS2000 the difference is 0.23+/-0.02 year/decade). Given the correction scheme is based on exactly this difference of apparent to ideal mean age, the results from Loeffel et al (2022) suggest that basing the correction scheme on the transient or timeslice simulation will make no difference. However, we did find in Loeffel et al (2022) a minor effect from the transient evolution of the chemical reactant species that are important for SF6-depleting chemistry, which thus would alter the relation of apparent to ideal age over time. This effect might be in particular important at higher altitudes. However, given the uncertainty in SF6 chemistry (see also below) and lack of observational constraints, it is not clear whether including this transient sink strength would overall improve the correction scheme.

We added the following sentences when introducing the simulation set-up in Section 3.1 (see line 193 ff in revised manuscript):

"*The setup is designed to investigate the effects of the SF6 sinks under constant climate conditions, and was chosen for this study to better isolate the effects of the SF6 sinks without the influence of model-dependent secular trends in circulation and composition. It was shown by Loeffel et al. (2022) that the transient evolution of circulation and composition only has minor influences on the the difference of ideal AoA to apparent AoA, so that a transient simulation would give similar results to the timeslice simulation in terms of the sink correction scheme.*"

The work is based on the chemical scheme of EMAC. Other models will likely give different SF6 loss rates and the range of lifetimes is given in the final section. What would be the implications is the shortest estimates of the SF6 lifetime were correct?

We agree with the reviewer that the chemical sinks of SF6 bear large uncertainty (as discussed in the paper in the concluding Section 6), and using a different model would certainly affect the sink correction scheme. The shortest lifetime estimates of around 500 – 1000 years (Reddmann et al., 2001; Ray et al., 2017) are by a factor 1/2 to 1/4 below the estimate from EMAC. For a lifetime around 500 years, if the parameterized effective lifetime would scale with the global stratospheric lifetime (which is not necessarily the case!), this would imply an effect on apparent age between a factor of 1.5 (for mean age = 2 years) to a factor of 2.4 (for mean age = 5 years). An implication might also be that the strong increase of apparent age already occurs for younger ideal ages, thus the age range in which the correction work well might be reduced. However, given the comparison to observational data (see Fig. 13), we do not find a clear indication for this to be the case.

To better acknowledge the overall large uncertainty in our knowledge on the SF6 sink, we expanded the concluding sentence of the paper as follows:

"*However, future work is needed to test and better constraint the parameters for the sink correction scheme, and to generally enhance our knowledge on the chemical sinks of SF6 in the atmosphere.*"

Line 46. 'tracer' (singular)

Done.

Line 47. 'particular'

Done.

Line 48. 'often-measured', 'has become increasingly clear'.

Done.

Line 70. 'next section. Here it would be better to give the explicit section number.

Done.

Line 147. 'constrained'.

Done.

Line 173. The caption of Figure 1 could do with some references for the data and model values.

Done, included references to model and observational reference time series.

Line 181. 10^6.

Done.

Line 231. MLS N2O is known to have a drift, especially in the older versions. Please clarify what version is use here (in GOZCARDS), whether it is subject to a drift and if so what is the impact on the AoA derived using it.

As documented by Froidevaux et al (2015), the Aura MLS N2O version 3.3 data record was used for the GOZCARD product. They also state that data was only used prior to 2013, because of a degradation in the N2O band, and that the GOZCARDS N2O product shows " insignificant drifts". Therefore, drifts are likely not a major contribution to the uncertainty derived from the N2O GOZCARDS product. Rather, the larger uncertainty is likely introduced via the assumed relationship between N2O and age used to derive mean age (described in Linz et al, 2017), as discussed already in our study (see lines 571 ff of revised paper).

Line 258. Figure 3 left panel. You might as well have a legend which labels the 5 lines explicitly.

Good idea, and done.

Line 285. Space before '+'.

Done.

Line 286. I think the partial lifetimes of the EMAC SF6 tracer should be stated more clearly and comprehensively (in Section 3.1). Here we are told 1900 years for stratosphere and mesospheric loss. (Assuming no loss in the troposphere that would be the total lifetime?). The conclusions (line 533) say 2100 for just the stratosphere (why just the stratosphere for this part?). These values may be consistent but it would be better to summarise up front what partial and overall lifetimes are produced by this version of EMAC.

Agreed, and thanks for pointing this out. Indeed in both instances the total middle atmospheric (stratospheric + mesospheric) lifetime was referred to, and in the conclusion a value from a different simulation was chosen by mistake. We corrected the value, and followed the advise to state the lifetime up front in Section 3.1 (see line 189 ff in revised paper):

*"The overall global lifetime of SF6 in the stratosphere and mesosphere is around 1900 years in the model (Loeffel et al., 2022). In general, the atmospheric SF6 lifetime is subject to large uncertainties, with recent estimates ranging from about 800 to 3000 years (Ray et al., 2017; Kovács et al., 2017; Kouznetsov et al., 2020), encompassing the SF6 lifetime in the EMAC model."*

Line 294 (and many other places) '3$^{rd}$-order'.

Done (and likewise 5$^{th}$-order).

Line 300. 'evaluated as well'.

Done.

Line 303. You are mixing US and UK spellings. (Also for color / colour). Also 'non-physical'.

Fixed.

Line 323. Need to rephrase, e.g. 'this spin up effect …. is not only an internal property of the model..'?

Thanks for the suggestion, and rephrased.

Line 340. 'lifetime'

Done.

Line 401. 'sections'.

Done.

Line 416. Degrees symbols in latitude range.

Done.

Line 437 (and line 464). Better to say 'previous section' or give the number.

Done.

Line 457. '21-year'.

Done.

Line 543. 'Plumb' is misplaced.

Done.

Note that all references cited here can be found in the originally submitted manuscript.